evolution/molecular biology

Gammaridae, paraphyletic, monophyletic, high-altitude adaptation, *atp8*, non-adaptive mutational pressure

**Author for correspondence:**
Hongtuo Fu
e-mail: fuht@ffrc.cn

†These authors contributed equally to the work.

# Disentangling the interplay of positive and negative selection forces that shaped mitochondrial genomes of *Gammarus pisinnus* and *Gammarus lacustris*

Shengming Sun[1,†], Ying Wu[1,†], Xianping Ge[1,2], Ivan Jakovlić[3], Jian Zhu[1,2], Shahid Mahboob[4,5], Khalid Abdullah Al-Ghanim[4], Fahad Al-Misned[4] and Hongtuo Fu[1]

[1]Wuxi Fisheries College, Nanjing Agricultural University, Wuxi 214081, People's Republic of China
[2]Agriculture Ministry Key Laboratory of Healthy Freshwater Aquaculture, Zhejiang Institute of Freshwater Fisheries, Huzhou 313001, People's Republic of China
[3]Bio-Transduction Lab, Wuhan, People's Republic of China
[4]Department of Zoology, College of Science, King Saud University, PO Box 2455, Riyadh-11451, Riyadh, Saudi Arabia
[5]Department of Zoology, GC University, Faisalabad, Pakistan

IJ, 0000-0002-2461-3712; SM, 0000-0003-4969-5387

We hypothesized that the mitogenome of *Gammarus lacustris* (GL), native to the Qinghai-Tibet Plateau, might exhibit genetic adaptations to the extreme environmental conditions associated with high altitudes (greater than 3000 m). To test this, we also sequenced the mitogenome of *Gammarus pisinnus* (GP), whose native range is close to the Tibetan plateau, but at a much lower altitude (200–1500 m). The two mitogenomes exhibited conserved mitochondrial architecture, but low identity between genes (55% *atp8* to 76.1% *cox1*). Standard (homogeneous) phylogenetic models resolved Gammaridae as paraphyletic, but 'heterogeneous' CAT-GTR model as monophyletic. In indirect support of our working hypothesis, GL, GP and *Gammarus fossarum* exhibit evidence of episodic diversifying selection within the studied Gammaroidea dataset. The mitogenome of GL generally evolves under a strong purifying selection, whereas GP evolves under directional (especially pronounced in *atp8*) and/or relaxed

selection. This is surprising, as GP does not inhabit a unique ecological niche compared to other gammarids. We propose that this rapid evolution of the GP mitogenome may be a reflection of its relatively recent speciation and heightened non-adaptive (putatively metabolic rate-driven) mutational pressures. To test these hypotheses, we urge sequencing mitogenomes of remaining *Gammarus* species populating the same geographical range as GP.

## 1. Background

The amphipod crustacean genus *Gammarus* Fabricius, 1775 (Senticaudata: Gammaroidea: Gammaridae) contains more than 200 described species [1], which makes it one of the most speciose crustacean genera [2]. These species are widely distributed throughout the Northern Hemisphere in various marine, brackish, freshwater and subterranean aquatic habitats [3]. *Gammarus lacustris* (GL) is a freshwater species with a nearly circumboreal range [1], widely distributed in alpine lakes and native to the Qinghai-Tibet Plateau (QTP) [3]. The QTP is the highest and largest plateau in the world, on average about 3000–4500 m in altitude [4]. As opposed to GL, *Gammarus pisinnus* (GP) inhabits rivers and lakes in a narrow geographical range comprising the plains south of the Taihang mountains, along the Weihe River in Shanxi Province [5], several hundred kilometres eastward from the eastern edge of the QTP, and at a much lower altitude of 200–1500 m [5]. The uplifting of the QTP dramatically changed the environmental conditions from an originally humid and warm climate to the currently dry and cold one [6]. Such ecological changes are expected to result either in the extinction of a species from the habitat or in major genetic and phenotypic adaptations to the new environment. Indeed, the QTP is typically inhabited by endemic species [7,8] highly adapted to the extreme environmental conditions, such as hypoxia, low temperatures and strong ultraviolet radiation [3,4,9–14]. Although amphipod crustaceans are the predominant macroscopic aquatic invertebrate in the QTP [15], specificities of the high-altitude adaption of crustaceans remain poorly understood. Furthermore, GL has been found at Tibetan localities exceeding elevations of 5000 m, which is currently the highest known altitudinal record for gammarids [8]. This exceptional ability of GL to inhabit extremely high altitudes, matched only by some molluscs among the aquatic invertebrates [7,8], is an indication that this species may be a very good model to investigate genetic adaptations to the high-altitude environment.

Owing to the abundance of mitochondria in cells, (mostly) maternal inheritance, the absence of introns, small genomic size (in metazoans), and an increasingly large set of available orthologous sequences, mitochondrial genomes (mitogenomes) have become a popular tool in population genetics [16], phylogenetics [17–20], diagnostics [20–22] and evolution [23–26]. Although long-considered to evolve under the purifying selection, there is increasing evidence that mitogenomes can undergo episodes of directional selection in response to shifts in physiological or environmental pressures [27]. The mitochondrial genome harbours genes involved in oxidative phosphorylation, and thus plays a central role in the energy production in the organism [28]. Due to low temperatures and hypoxia, life at high altitudes is expected to generate unique evolutionary pressures on the energy management system of organisms [9,10]. However, high-altitude adaptation can occur on different metabolic levels, not all of which include the mitochondrial metabolism [29,30], so mitochondrial genome may reflect only a small fragment of these adaptive changes. Despite this limitation, a number of studies have detected signals of directional selection in the mitogenomes of animals living at high elevations [30–32], including both terrestrial [11,12,29,33,34] and aquatic [35] Tibetan fauna. Therefore, our working hypothesis was that the speciation of GL (but not that of GP) was accompanied by mitogenomic adaptations to the extreme environment of the Tibetan plateau. To test this hypothesis, we sequenced the complete mitochondrial genomes of GL and GP, conducted comparative mitogenomic and phylogenomic analyses using a set of available gammarid mitogenomes, and studied evolutionary signals imprinted upon the two mitogenomes.

## 2. Results

### 2.1. Characteristics and architecture of the two mitogenomes

The length of complete mitochondrial genomes was: 15 907 bp in GP, 15 333 bp in GL. Both possessed the standard 13 protein-coding genes (PCGs), two rRNA genes (16S and 12S), 22 tRNA genes and a control region. Both mitogenomes exhibited an identical, highly conserved, gene order and strand distribution (table 1). The genes of the two mitogenomes also exhibited very similar sizes, partially shared start/

**Table 1.** Organization of the mitogenomes of *Gammarus lacustris* and *Gammarus pisinnus*. Data are presented as *G. lacustris*/*G. pisinnus*. IGN is intergenic region, where a negative number represents an overlap. Sizes are given in bp (base pairs). CR is the putative control region.

| gene | position | | size | IGN | codon | | strand | identity |
|---|---|---|---|---|---|---|---|---|
| | from | to | bp | bp | start | stop | | % |
| *trnY* | 1/1 | 61/62 | 61/62 | | | | -/- | 62.9 |
| *trnQ* | 58/59 | 113/116 | 56/58 | -4/-4 | | | -/- | 72.41 |
| *trnC* | 369/335 | 426/391 | 58/57 | 255/218 | | | -/- | 62.07 |
| *trnI* | 440/583 | 500/642 | 61/60 | 13/191 | | | +/+ | 80.33 |
| *trnM* | 502/646 | 562/705 | 61/60 | 1/3 | | | +/+ | 78.69 |
| *nad2* | 563/706 | 1544/1687 | 982/982 | | TTG/TTG | T-/T- | +/+ | 62.23 |
| *trnW* | 1545/1688 | 1604/1748 | 60/61 | | | | +/+ | 86.89 |
| *trnG* | 1606/1749 | 1665/1808 | 60/60 | 1/- | | | +/+ | 83.33 |
| *cox1* | 1666/1809 | 3202/3345 | 1537/1537 | | ATC/ATA | T-/T- | +/+ | 76.06 |
| *trnL2* | 3203/3346 | 3262/3405 | 60/60 | | | | +/+ | 85 |
| *cox2* | 3263/3406 | 3941/4084 | 679/679 | | TTG/ATT | T-/T- | +/+ | 72.31 |
| *trnK* | 3942/4085 | 4000/4142 | 59/58 | | | | +/+ | 88.14 |
| *trnD* | 4001/4143 | 4061/4203 | 61/61 | | | | +/+ | 86.89 |
| *atp8* | 4062/4204 | 4220/4359 | 159/156 | | ATT/ATA | TAA/TAA | +/+ | 55 |
| *atp6* | 4214/4353 | 4885/5024 | 672/672 | -7/-7 | ATG/TTG | TAA/TAA | +/+ | 66.67 |
| *cox3* | 4885/5024 | 5670/5809 | 786/786 | -1/-1 | ATG/ATG | TAA/TAA | +/+ | 73.03 |
| *nad3* | 5673/5811 | 6026/6164 | 354/354 | 2/1 | ATG/ATG | TAG/TAG | +/+ | 64.41 |
| *trnA* | 6025/6163 | 6084/6221 | 60/59 | -2/-2 | | | +/+ | 70 |
| *trnS1* | 6084/6221 | 6135/6271 | 52/51 | -1/-1 | | | +/+ | 80.77 |
| *trnN* | 6138/6274 | 6198/6334 | 61/61 | 2/2 | | | +/+ | 73.77 |
| *trnE* | 6196/6332 | 6257/6395 | 62/64 | -3/-3 | | | +/+ | 85.94 |

(Continued.)

**Table 1.** (*Continued.*)

| gene | position from | to | size bp | IGN bp | codon start | stop | strand | identity % |
|---|---|---|---|---|---|---|---|---|
| trnR | 6252/6390 | 6311/6450 | 60/61 | −6/−6 | | | +/+ | 77.05 |
| trnF | 6310/6449 | 6369/6507 | 60/59 | −2/−2 | | | −/− | 73.33 |
| nad5 | 6370/6508 | 8082/8226 | 1713/1719 | | GTG/TTG | TAA/TAA | −/− | 60.03 |
| trnH | 8083/8227 | 8141/8286 | 59/60 | | | | −/− | 80 |
| nad4 | 8142/8287 | 9456/9601 | 1315/1315 | | ATG/TTG | T−/T− | −/− | 67.76 |
| nad4 L | 9450/9595 | 9746/9888 | 297/294 | −7/−7 | ATA/ATG | TAA/TAG | −/− | 61.95 |
| trnT | 9747/9891 | 9807/9950 | 61/60 | −/2 | | | +/+ | 72.13 |
| trnP | 9808/9950 | 9867/10013 | 60/64 | −/−1 | | | −/− | 74.24 |
| nad6 | 9870/10016 | 10373/10516 | 504/501 | 2/2 | ATG/ATG | TAA/TAA | +/+ | 55.56 |
| cytb | 10373/10516 | 11507/11652 | 1135/1137 | −1/−1 | ATG/ATG | T−/TAA | +/+ | 66.14 |
| trnS2 | 11508/11651 | 11568/11712 | 61/62 | −/−2 | | | +/+ | 64.52 |
| nad1 | 11594/11784 | 12505/12716 | 912/933 | 25/71 | TTG/TTG | TAA/TAA | −/− | 67.63 |
| trnL1 | 12506/12717 | 12568/12775 | 63/59 | | | | −/− | 66.67 |
| rrnL | 12569/12776 | 13548/13763 | 980/988 | | | | −/− | 71.72 |
| trnV | 13549/13764 | 13602/13814 | 54/51 | | | | −/− | 74.07 |
| rrnS | 13603/13815 | 14352/14531 | 750/717 | | | | −/− | 65.3 |
| CR | 14353/14532 | 15333/15907 | 981/1376 | | | | | |

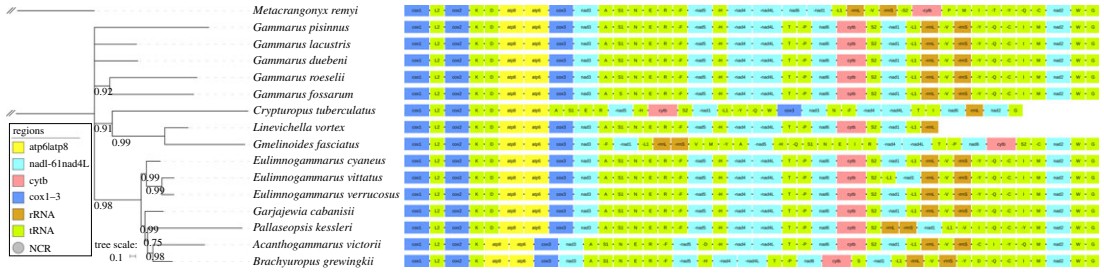

**Figure 1.** Mitochondrial phylogenomics and gene orders of the superfamily Gammaroidea. Phylogram was inferred using nucleotide sequences of all mitochondrial genes (PCG + rRNA + tRNA) and CAT-GTR model implemented in PhyloBayes. *Metacrangonyx remyi* is the outgroup (branch cropped). Scale bar corresponds to the estimated number of substitutions per site. Posterior probability support values less than 1.0 are shown next to nodes, and GenBank accession numbers are available in the electronic supplementary material, Additional file S1. Gene orders are shown to the right, with the legend included in the figure.

stop codons, and very similar intergenic regions (IGR) and overlaps. Size-wise, there were a few exceptions: *rrnS* gene was larger in GL than in GP (750 versus 717 bp), whereas *nad1* was much shorter in GL (912 bp) than in GP (933 bp); GP had a much larger IGR between *trnI* and *trnC* (191 versus 13 bp) and a much larger (putative) control region (1376 versus 981 bp). Several start codons were found, ATG, TTG, ATA, ATT, GTG and ATC, and most genes used the standard TAA stop codon (including an abbreviated form of it T−). An exception was *nad3* of both species and *nad4 L* of GP, which used TAG. GP had the A + T content of 70% and GL of 64.3%. None of these architectural features are novel or unique among the Gammaroidea and Senticaudata (electronic supplementary material, Additional file S1).

AT and GC skew values of complete mitogenomes were −0.068 and −0.31 (respectively) for GP; and −0.014 and −0.263 for GL. As regards individual genes, all PCGs and rRNAs exhibited predominantly negative AT skews, but tRNAs (concatenated) had positive AT skews (electronic supplementary material, Additional file S1). Several exceptions (inverted skews) were observed: the two rRNA genes in the GP mitogenome exhibited inverted AT skew values (*rrnL* = 0.004 and *rrnS* = 0.115); *rrnS* in *G. fossarum* and *Gmelinoides fasciatus* mitogenomes (0.016 and 0.017, respectively); *atp8* of *Eulimnogammarus vittatus* (0.008) and *G. roeselii* (0.036); and tRNAs of *Stygobromus indentatus* (−0.017).

However, although the architecture of the two mitogenomes was very similar, gene identity values were relatively low: 72% on average, ranging from 55% (*atp8*) to 88% (*trnK*). Among the 13 PCGs, the average identity was even lower (65.3%), with *cox1* exhibiting the highest value of 76.1%. As regards gene families, *cox* family (*cox1–3*) exhibited the highest average value (73.8%), followed by the *nad* family (62.8%) and finally the *atp* family (60.84%). Although tRNA remoulding has been reported in this group of crustaceans before [36], we did not find evidence for such evolutionary events in the two newly sequenced species.

## 2.2. Phylogeny and gene order

Gene order was perfectly conserved in the entire Gammaridae family, including the two newly sequenced species (figure 1). Different datasets and algorithms produced notable topological instability. Apart from the CAT-GTR analysis of the PCGRT dataset (figure 1), other analyses produced deeply paraphyletic family Gammaridae, mostly with three or even four Baikal lake families (Acanthogammaridae, Pallaseidae, Eulimnogammaridae + Micruropodidae) nested within the clade (electronic supplementary material, Additional file S2). CAT-GTR model resolved Gammaridae as monophyletic, notwithstanding several nodes in the phylogram that exhibited unresolved polyphyletic relationships. Apart from Acanthogammaridae, which were rendered paraphyletic by the nested Pallaseidae (*Pallaseopsis kessleri*), family, the remaining Gammaroidea families (Micruropodidae and Eulimnogammaridae) were also monophyletic (taxonomic details available in electronic supplementary material, Additional files S1 and S2).

## 2.3. Direction and magnitude of selective pressures

The rate of non-synonymous (dN) to synonymous (dS) substitutions, $\omega = dN/dS$, is a commonly used indicator for measuring the direction and magnitude of selective pressure on genes, where $\omega < 1$

indicates purifying selection, $\omega = 1$ indicates neutral evolution and $\omega > 1$ indicates directional (or positive) selection [37]. In the dataset comprised of 13 concatenated PCGs, the number of segregating (variable) sites was 1283, total number of observed mutations was 2061, among which 1865 were synonymous, and 196 were non-synonymous.

First, we used aBSREL, which allows a global test for positive selection in the entire dataset [38]. Using the concatenated 13 PCGs dataset with all 15 species (apart from the outgroup *M. remyi*) selected as test branches, we found statistically significant evidence ($p$-value corrected for multiple testing $\leq 0.05$) of episodic diversifying selection in two species: *Gammarus fossarum* ($p = 0.0026$), and the newly sequenced GP ($p = 0.0095$). The corrected $p$-value of GL was also very close to the significance threshold ($p = 0.0658$; the uncorrected $p$-value was 0.05). For the global test, $p$-values at each branch must be corrected for multiple testing, which decreases the selection detection power of this exploratory approach [38], so we then used the same tool to conduct independent tests for GL and GP (selected as the test branch), and the remaining species as reference branches. Results of these two tests were as expected, within the context of the global test: evidence of episodic diversifying selection was identified in both species ($p$-values: GL = 0.0052, GP = 0.0007).

Using the BUSTED tool, which 'provides a gene-wide (not site-specific) test for positive selection by asking whether a gene has experienced positive selection at at least one site on at least one branch' [39], we found evidence of gene-wide episodic diversifying selection in both species (independently selected as test branches): $p = 0.048$ for GL, and $p = 0.001$ for GP.

Relaxation of the efficiency or intensity of natural selection can result in a loss of function, but also drive evolutionary innovation [40], so we also explored the direction and magnitude of selection pressures on GL and GP using the RELAX algorithm, which asks whether the strength of natural selection has been relaxed or intensified along a specified set of test branches. While not suitable for explicitly testing for positive selection, this method is useful for identifying trends and/or shifts in the stringency of natural selection on a given locus [40]. A significant result of $K > 1$ indicates that selection strength has been intensified along the test branches, and a significant result of $K < 1$ indicates that selection strength has been relaxed along the test branches. Both analyses produced statistically significant results, but the tests indicate selection *intensification* in GL, and selection *relaxation* in GP (table 2).

To get a closer perspective and test which genes may be evolving under unique selection pressures, we divided the dataset into individual genes and conducted another round of analyses on all 13 individual genes (table 2). We did not find statistically significant evidence of directional selection (aBSREL + BUSTED) in any of the GL genes, but $p$-values indicate that *cox3* may be undergoing higher levels of directional selection pressures than other genes. RELAX analyses revealed that all GL genes except *nad4 L* (significant) and *nad6* (non-significant) exhibited intensified selection (six significant: *atp6, cox1, cox3, cytb, nad3* and *nad5*). In GP, *atp8* (aBSREL + BUSTED) and *nad3* (aBSREL) exhibited statistically significant evidence of directional selection. Most genes exhibited different levels of relaxed selection pressures (statistically significant in *cytb, nad4* and *nad6*), with only *cox1, nad4 L* and *nad5* exhibiting (non-significantly) intensified selection pressures.

To further explore the selection signals using a per-site-based approach in selected genes of interest (table 3), we relied on the FEL algorithm, which uses a maximum-likelihood approach to infer dN and dS substitution rates on a per-site basis for a given coding alignment and corresponding phylogeny [37]. First, we tested the entire dataset, and then genes for which other analyses (table 2) indicated that they may be evolving under unique selection pressures (significance threshold was set at $p < 0.05$ for all analyses). For the entire dataset (13PCG), the sites exhibiting predominantly negative (purifying) selection by far outnumbered the sites exhibiting predominantly positive (diversifying) selection (897 versus 17, respectively) in GL, but the situation was reversed in GP, where sites exhibiting pervasive positive/diversifying selection by far outnumbered the sites exhibiting pervasive negative/purifying selection (619 versus 0, respectively). This was reflected in individual genes as well, where all three GP genes (*atp8, cox3* and *nad3*) exhibited only positive selection. In GL, *nad4 L* and *nad6* also exhibited only positive selection, whereas *cox3* exhibited only negative selection.

Finally, to better examine the signal of episodic diversifying selection in *Gammarus fossarum* observed in the exploratory aBSREL test ($p = 0.0026$), we conducted a few additional analyses with this species selected as the test branch. aBSREL ($p = 0.0002$) and BUSTED found evidence ($p = 0.000$) of episodic diversifying selection. RELAX test for selection relaxation ($K = 0.84$) was also significant ($p = 0.001$, LR = 11.52), which was supported by the FEL analysis, where sites exhibiting pervasive

**Table 2.** Selection signals in the mitogenomes of *Gammarus lacustris* (GL) and *Gammarus pisinnus* (GP) inferred using BUSTED, aBSREL and RELAX algorithms. 13PCG is the dataset comprising all 13 concatenated (non-partitioned) protein-coding genes. Sites column shows the number of sites in the alignment. $K$ column: a statistically significant $K > 1$ indicates that selection strength has been intensified, and $K < 1$ that selection strength has been relaxed. LR is likelihood ratio, $p$ is $p$-value and $D$ indicates the direction of selection pressure change: intensified ($I$) or relaxed ($R$), where * highlights a statistically significant ($p < 0.05$) result.

| data | sites | Gammarus lacustris (GL) | | | | | | Gammarus pisinnus (GP) | | | | | |
| | | BUSTED | aBSREL | RELAX | | | | BUSTED | aBSREL | RELAX | | | |
| | | p-value | p-value | K | LR | p-value | D | p-value | p-value | K | LR | p-value | D |
| 13PCG | 3743 | 0.024* | 0.005* | 1.21 | 11.54 | 0.001 | I* | 0.001* | 0.010* | 0.63 | 56.39 | 0.000 | R* |
| atp6 | 223 | 1.000 | 1.000 | 14.80 | 7.48 | 0.006 | I* | 0.487 | 1.000 | 0.68 | 1.51 | 0.219 | R |
| atp8 | 58 | 1.000 | 1.000 | 1.02 | 0.02 | 0.897 | I | 0.000* | 0.004* | 0.13 | 1.14 | 0.286 | R |
| cox1 | 527 | 1.000 | 1.000 | 1.87 | 12.31 | 0.000 | I* | 1.000 | 1.000 | 1.00 | −0.38 | 1.000 | I |
| cox2 | 231 | 1.000 | 1.000 | 1.60 | 2.76 | 0.097 | I | 1.000 | 1.000 | 1.00 | 0.00 | 1.000 | R |
| cox3 | 262 | 0.345 | 0.087 | 2.24 | 8.05 | 0.005 | I* | 0.327 | 0.144 | 0.45 | −6.96 | 1.000 | R |
| cytb | 380 | 1.000 | 0.168 | 2.37 | 6.63 | 0.010 | I* | 1.000 | 1.000 | 0.46 | 7.91 | 0.005 | R* |
| nad1 | 313 | 1.000 | 1.000 | 1.18 | 0.90 | 0.343 | I | 0.749 | 1.000 | 0.77 | 3.24 | 0.072 | R |
| nad2 | 330 | 0.718 | 1.000 | 1.01 | 0.37 | 0.543 | I | 0.562 | 0.431 | 0.84 | 0.42 | 0.516 | R |
| nad3 | 119 | 1.000 | 1.000 | 7.91 | 6.98 | 0.008 | I* | 0.049* | 0.454 | 0.64 | 2.28 | 0.131 | R |
| nad4 | 447 | 1.000 | 1.000 | 1.04 | 0.40 | 0.530 | I | 1.000 | 1.000 | 0.58 | 7.75 | 0.005 | R* |
| nad4 L | 98 | 1.000 | 1.000 | 0.36 | 4.98 | 0.026 | R* | 1.000 | 1.000 | 2.05 | 2.61 | 0.106 | I |
| nad5 | 579 | 0.741 | 0.385 | 1.38 | 4.23 | 0.040 | I* | 0.460 | 0.500 | 2.15 | −45.82 | 1.000 | I |
| nad6 | 176 | 0.766 | 0.500 | 0.95 | −0.09 | 1.000 | R | 1.000 | 1.000 | 0.49 | 4.41 | 0.036 | R* |

**Table 3.** Selection signals in the selected genes of mitogenomes of *Gammarus lacustris* (GL) and *Gammarus pisinnus* (GP) inferred using a per-site-based method—FEL. 13PCG is the dataset comprising all 13 concatenated (non-partitioned) protein-coding genes. Sites column shows the number of sites in the alignment. + indicates the number of sites evolving under significant (threshold set at $p < 0.05$) pervasive positive/diversifying selection, and − under pervasive negative/purifying selection.

| data | sites | GL | | GP | |
|---|---|---|---|---|---|
| | | + | − | + | − |
| 13PCG | 3743 | 17 | 897 | 619 | 0 |
| atp8 | 58 | | | 18 | 0 |
| cox3 | 262 | 0 | 78 | 37 | 0 |
| nad3 | 119 | | | 17 | 0 |
| nad4 L | 98 | 19 | 0 | | |
| nad6 | 176 | 14 | 0 | | |

positive/diversifying selection by far outnumbered the sites exhibiting pervasive negative/purifying selection (346/0 respectively; $p < 0.05$).

# 3. Discussion

To test the hypothesis that mitochondrial genome of GL may exhibit imprints of genetic adaptation to a high-altitude environment of the Tibetan plateau, we compared it to the mitogenome of a congeneric species inhabiting lower altitudes, GP, as well as the dataset comprising all currently available Gammaroidea mitogenomes. To achieve this, we sequenced the complete mitochondrial genomes of these two species, conducted comparative mitogenomic and phylogenetic analyses, and applied a number of different tests to study the evolutionary forces that shaped these mitogenomes.

## 3.1. General comparative mitogenomics

Both mitogenomes exhibited a fairly conserved architecture, standard for Gammaroidea and Senticaudata (figure 1; electronic supplementary material, Additional file S1). A notable exception was the truncated *nad1* gene in GL (912 bp), also shared by *Gammarus duebeni* (913 bp), whereas most other Senticaudata (including GP) exhibited a larger size (924–933 bp). As the 3′ of this gene is poorly conserved in Senticaudata, this is an indication of its limited importance for the functionality of this gene, so it is unlikely that the truncation was adaptive, and more likely that it is merely a consequence of a non-adaptive random mutational event introducing a stop codon into a non-essential segment of the gene.

## 3.2. Strand asymmetry and skews

Strand asymmetry is a relatively common feature of organellar genomes [41–43], including those of gammarids [36], but it can obstruct phylogenetic reconstruction and produce gravely misleading phylogenetic artefacts (long-branch attraction) in cases when unrelated taxa exhibit inverted skews [44,45]. GP exhibited the lowest AT skew value among the available Gammaroidea mitogenomes (−0.068) and a low GC skew value of −0.31, whereas GL exhibited a comparatively standard AT (−0.014) and GC (−0.263) skew values. GP also exhibited the highest A + T content (70%) among the Gammaroidea mitogenomes, whereas GL had a comparatively low A + T content of 64.3%. This is an indication that the evolutionary forces (adaptive or non-adaptive) that shaped the mitogenome of GP may have differed from those of other gammarids. Whereas most PCGs had negative GC skews, *nad1*, *nad4*, *nad4 L*, *nad5* and all RNA genes had positive GC skews. Importantly, these genes are all encoded on the minus strand, which is in agreement with the hypothesis that strand asymmetry is caused by hydrolytic deamination of bases in the leading DNA strand when it is single stranded, i.e. during replication and transcription [43]. The only individual outlier in relation to this pattern was the GC skew value of tRNAs of *S. indentatus* (−0.017) (also an outlier in the AT skew). Disregarding the

minor exception of GP, gammarids exhibit relatively consistent skews, so we can tentatively conclude that there are no indications that mitogenomes are not a suitable tool for reconstructing the phylogeny of gammarids.

## 3.3. Phylogeny and gene order

In congruence with the topologies obtained using the standard (homogeneous) models in combination with data partitioning (electronic supplementary material, Additional file S2), paraphyly of Gammaridae has been reported in a relatively recent multi-gene-based revision of the Gammaroidea phylogeny [46], and the split of the Baikal lake Gammaroidea into two clades has also been reported by several studies [46–48]. However, using an almost complete mitogenomic dataset (PCGRT) in combination with a 'heterogeneous' CAT-GTR model, we managed to obtain monophyletic (polytomy notwithstanding) Gammaridae. This supports a previous observation that CAT-GTR model appears to be more efficient at dealing with compositional heterogeneity than data partitioning [45]. Although we tentatively accepted this as the correct topology, this reasoning is somewhat circular, so the monophyly of this family should be corroborated using additional datasets and data types before this issue can be declared as resolved. The close relationship of GL and *G. duebeni* in all of our topologies (polyphyly in CAT-GTR notwithstanding) is in disagreement with the findings of Hou & Sket [46], and rather unexpected, as *G. duebeni* is distributed throughout the intertidal zone of the North Atlantic region [49]. It might be merely an artefact caused by the poor sampling of gammarid mitogenomes, resulting in unresolved polyphyletic relationships, or even particular evolutionary pressures that have shaped the mitogenome of GL. A much denser sampling of closely related gammarid mitogenomes would be needed to test these hypotheses.

While the two newly sequenced mitogenomes exhibited perfectly conserved gene orders (among the Gammaridae), a representative of the Baikalian family Micruropodidae, *Gmelinoides fasciatus*, exhibited a highly rearranged gene order (figure 1), which is very unusual among the Gammaroidea. The remaining two available Micruropodidae mitogenomes are incomplete, so it is impossible to infer conclusions with confidence, but *Linevichella vortex* appears to possess a conserved gammarid gene order, whereas *Crypturopus tuberculatus* also seems to possess a highly rearranged and unique gene order (figure 1). This contrast between the highly conserved architecture in the rest of the Gammaroidea dataset and highly volatile gene order in some Micruropodidae might offer strong evidence in support of the hypothesis that the evolution of mitogenomic architecture is highly discontinuous [23]. Therefore, it would be of greatest interest to sequence further mitogenomes belonging to the Micruropodidae and the entire Baikal group 2 of gammarids, as defined by Hou & Sket [46].

## 3.4. Selective pressures

Given the central role of the mitochondrial genome in the energy production in animals [31], we hypothesized that the mitogenome of GL may exhibit signs of adaptation (directional selection) to life at high altitudes. In agreement with our hypothesis, we did find evidence for directional selection in GL using two different algorithms (aBSREL and BUSTED). Although, results of RELAX and FEL algorithms indicate that signals of purifying selection outweigh the signals of directional selection in the mitogenome of this species, this is expected, as purifying selection is the predominant force driving the evolution of mitogenomes, and directional selection signals are usually constrained to specific functional sites within genes [30,50,51].

It should be noted here that GL possesses relatively high passive dispersal potential, largely due to its ability to be dispersed short distances via waterfowl, which results in a broad distribution range of this species [1,8]. This makes it unique both among other Tibetan life forms, which tend to be endemic [7], and other freshwater *Gammarus* species, which generally have weak dispersal potential [5]. Therefore, we should not exclude a possibility that Tibetan GL populations may be receiving a limited influx of migrants from other locations, which may weaken the directional selection signal. As this species is generally well-adapted to life in alpine lakes, these migrants may be able to survive in the extreme environment of the QTP. However, molecular phylogeographic patterns indicate (due to limited data, this is not strongly supported) that gammarid fauna of the Tibetan plateau is probably the result of a single recent colonization event, and that genetic admixture occurs only among different Tibetan populations [8]. Therefore, it is not a likely scenario. Furthermore, we would still expect the migrants to exhibit adaptations to life at high altitudes. Future studies may attempt to sequence mitogenomes

from lowland GL populations, and ideally even those inhabiting localities exceeding elevations of 5000 m [8], and conduct comparative analyses on the entire dataset.

Importantly, and unexpectedly, all tests indicate the existence of an even stronger signal of positive/diversifying selection, as well as overall significantly relaxed purifying selection, in the mitogenome of the other newly sequenced species, GP. This fast evolution of GP is also a likely underlying reason for the surprisingly low (for congenerics) [51,52] gene identity values between GL and GP. The evidence for relaxation of functional evolutionary constraints (and/or positive selection) was reported in the transcriptome of a subterranean *Gammarus* species [53], which can be explained by adaptation to a radically different (subterranean) environment. However, this finding is surprising for GP, as this species does not inhabit a unique ecological niche compared to other gammarids.

As mentioned above, most freshwater *Gammarus* species generally have weak dispersal potential, so their phylogeny is strongly influenced by geological barriers, which results in the existence of many endemic species, often with altitude-specific distribution ranges [5]. We hypothesize that the speciation of GP was most likely a result of fragmentation of aquatic habitats and changes in the environment that accompanied the rapid uplifting of the Taihang range from a vast plain that pre-dated it [5]. This process opened new ecological niches, where mutant phenotypes could be advantageous. For example, it has been proposed that uplifting of the Lüliang and Taihang ranges promoted several speciation events in gammarids, resulting in the existence of several (putative) gammarid species (including GP) in a relatively narrow geographical range [5]. Such scenario may explain the observed rapid evolution of the GP mitogenome, as well as the proposed heightened speciation intensity in these mountain ranges.

Moreover, this may also explain the much more pronounced directional selection pressure signal in the mitogenome of GP, compared to that of GL. The geological ages of the two habitats are very different; the uplift of the Tibetan plateau began during the Upper Cretaceous, some 70 Ma (million-years-ago), whereas the uplift of the Taihang mountain range occurred during the Miocene (23 to 5 Ma) [3,5]. Therefore, it is possible that GL underwent the adaptation to the new environment much earlier than GP. Such scenario would allow the purifying selection to swamp the signal of episodic positive evolution in GL, but not in GP. Note that this appears incongruent with the proposal mentioned above, that Tibetan GL populations are the result of a single recent colonization event [8], but the authors did not propose the exact time frame for 'recent', so we cannot reject it with certainty.

Furthermore, as accelerated mitochondrial evolution can be driven by directional adaptive selection or by non-adaptive mutational pressures [14,27,45], we should not exclude a possibility that the comparatively strong signal in GP could be a combination of both. The exceptionally high GC skew value of the GP mitogenome may be an indirect evidence for this, as base composition skews are believed to be mostly caused by non-adaptive mutational pressures, usually driven by architectural rearrangements or heightened metabolic rates [43–45]. Given the highly conserved architecture of the gammarid mitogenomes, we can probably exclude architectural rearrangements as the underlying cause, and instead propose that it may be associated with a (hypothetical) comparatively high metabolic rate in the GP. Although it is not excessive to speculate that species living in moderate climate environments would exhibit higher metabolic rates than species inhabiting extremely cold environment of the Tibetan plateau, this hypothesis would first have to be tested. Specific questions that remain open include: do other species described in the same geographical range [5] exhibit heightened evolutionary rates?; and do they exhibit higher evolutionary rates than *Gammarus* species inhabiting other moderate environments? Within the studied dataset, three (out of five) *Gammarus* species exhibited significant signals of episodic diversifying/relaxed selection: GL, GP and *G. fossarum*. This is an indication that some *Gammarus* species indeed do exhibit comparatively accelerated mitochondrial evolution rates in comparison to other Senticaudata. As these questions warrant further studies, it would be very interesting to sequence the mitogenomes of the remaining species described in that geographical range [5] and test the hypothesis that these mitogenomes should also exhibit comparatively high evolutionary rates within the entire *Gammarus* dataset. These molecular data would also allow us to further test the extent of isolation and gene flow between these proposed species.

In agreement with results obtained using the concatenated dataset, individual genes in the mitogenome of GL predominantly exhibited varying levels of purifying selection, whereas that of GP mostly exhibited relaxed selection. However, there were some outliers: *nad4 L* and *nad6* of GL exhibited relaxed selection pressure (only *nad4 L* significantly), whereas *cox1*, *nad4 L* and *nad5* of GP exhibited intensified selection pressure (all non-significant). Some of these outliers are expected, as *cox1* is generally the most conserved gene in animal mitogenomes [51,52,54], which includes both

other gammarids (Baikal lake) [36] and other Tibetan animals [31]. Our comparative analyses also indicate relatively high levels of purifying selection in the *cox* gene family of the two species (73–76% interspecific identity), and somewhat relaxed (or positive) selection in the *nad* family (55–68% identity). *nad4 L* is a very small gene that often overlaps with other genes (*nad4*) in many crustacean mitogenomes [17], which is an indication of reduced mutational constraints [55]; and *nad6* is often among the fastest-evolving mitochondrial genes [56,57], and sometimes even altogether missing from them [58]. Although this may explain the outliers in the GL mitogenome, it remains unclear why *nad4 L* and *nad5* of GP exhibited intensified selection pressure. As both results were non-significant, it may be just a statistical fluke.

A surprising finding was that, among the identified genes that may be especially strongly affected by selective pressures, GP *atp8* exhibited exceptionally strong evidence of directional selection. This is intriguing, as *atp8* is a very small, 'dispensable' gene, often missing from metazoan mitogenomes, and usually evolving under highly relaxed selection pressures [58]. Combined with the absence of evidence for a relaxation of selection constraints and very low similarity to the GL orthologue, we can conclude that our data indicate that *atp8* is indeed evolving under exceptionally strong directional selection in GP, which warrants further functional studies.

# 4. Conclusion

We identified the existence of directional evolution signal in the mitogenome of GL, which indirectly supports our working hypothesis, but a surprising finding is that the signal of relaxed/directional selective pressures is much stronger in the mitogenome of GP. This is surprising, as GP does not inhabit a unique ecological niche compared to other gammarids. We hypothesize that the speciation of GP was most likely a result of fragmentation of aquatic habitats and changes in the environment that accompanied the rapid uplifting of the Taihang range from a vast plain that pre-dated it, which opened new ecological niches where mutant phenotypes could be advantageous. Such scenario may explain the observed rapid evolution of the GP mitogenome, as well as the proposed heightened speciation intensity in these mountain ranges. We hypothesize that much older age of the Tibetan plateau may have allowed the purifying selection to swamp the signal of episodic positive evolution in GL, whereas the relative recency of its speciation may explain the stronger signal in GP. We propose that this hypothesis can be tested by sequencing the mitogenomes of the remaining six (proposed) species populating the same narrow geographical range. As all of these species are believed to be a result of these same geological events, their mitogenomes should exhibit similar evolutionary footprints. Furthermore, as we found indications that some *Gammarus* species exhibit comparatively accelerated mitochondrial evolution rates in comparison to other Senticaudata, we propose that future studies should not exclude the existence of unusually strong non-adaptive mutational pressures in their mitogenomes.

# 5. Methods

## 5.1. Sampling

GL samples were collected at the altitude of *ca* 3400 m, in the Cuomujiri lake (92°09′–98°47′ N, 26°52′–30°40′ E) near the Nyingchi city in Tibet (figure 2). GP samples were collected at the altitude of *ca* 510 m, from the Yellow River near the Xian City, Shanxi province, China (34°16′ N, 108°54′ E). Both samplings were conducted in August 2017 using fine-meshed hand nets. Approximately 50 specimens were collected at each site, immediately preserved in 95% ethanol in the field, and long-term stored at −20°C for later use. Samples were morphologically identified under the light microscope as described before [3,5]. As gammarids are non-protected invertebrates, no permits were needed for sampling and sample processing.

## 5.2. DNA extraction, amplification and sequencing

These steps were conducted as described before [23,57,59], so only an overview is provided. DNA was isolated from a complete single specimen (AidLab DNA extraction kit, AidLab Biotechnologies, Beijing, China) of each species after rinsing them in distilled water. Conserved fragments of several genes were amplified and sequenced using degenerate primer pairs (electronic supplementary

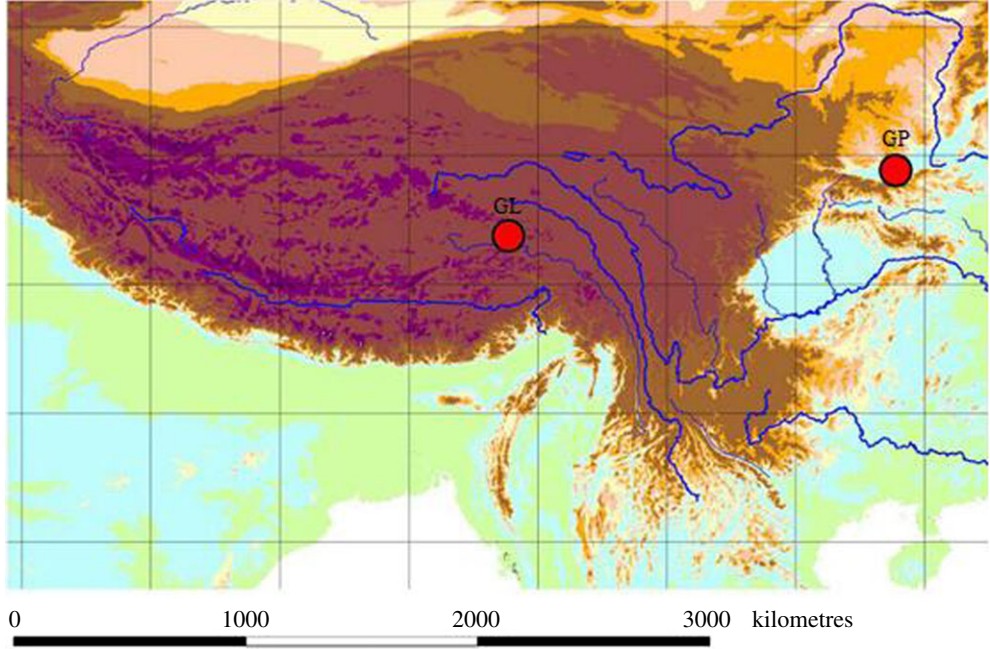

**Figure 2.** Approximate sampling localities for *Gammarus pisinnus* (GP) and *Gammarus lacustris* (GL).

material, Additional file S3) designed using available gammarid mitogenomes as templates. These gene fragments were then used to design specific primers (overlapping by approx. 100 bp) for amplification and sequencing of the whole mitogenome. PCR reaction mixture (20 μl): 7.4 μl ddH$_2$O, 10 μl 2 × PCR buffer (Takara, Dalian, China), 0.6 μl of each primer, 0.4 μl rTaq polymerase (250 U, Takara) and 1 μl DNA template. Amplification: 98°C/2 min, 40 cycles of 98°C/10 s, 50°C/15 s, 68°C 1 min kb$^{-1}$ and finally 68°C/10 min. PCR products were sequenced bi-directionally using the Sanger method and the same set of primers.

## 5.3. Sequence assembly and analysis

These steps, assembly, annotation and comparative analyses, were also conducted following the methodology described before in more detail [23,57,59]. Briefly, sequenced fragments were quality-proofed by visually inspecting the electropherograms and queried against the GenBank using BLAST to confirm that the amplicon is the target sequence. During the manual assembly, conducted with the help of DNAstar v. 7.1 [60], we made sure that the overlaps were identical, mitogenome circular, and that no mitochondrial DNA copies (numts) [61] were incorporated. ORFs for PCGs were located using DNAstar, and manually fine-tuned via a comparison with available gammarid orthologues using BLAST and BLASTx. tRNAscan [62] and ARWEN [63] were used to identify tRNAs. PhyloSuite [64] was used to parse and extract mitogenomes annotated in Microsoft Word documents, as well as to create the files for submission to GenBank and generate comparative mitogenomic architecture tables. Following the arguments put forward before [65], instead of referring to strands as heavy and light, we used plus (+) and minus (−) to denominate the strands encoding a majority and minority of genes, respectively. The mitogenomes are available from the GenBank repository under the accession numbers MK354235 (GL) and MK354236 (GP).

## 5.4. Comparative mitogenomic and phylogenetic analyses

To minimize the impact of compositional heterogeneity, for phylogenetic and selection pressure analyses we used only all 13 mitogenomes belonging to the superfamily Gammaroidea available in the GenBank, plus *Metacrangonyx remyi* (Senticaudata: Hadzioidea) [66] as the outgroup. As some Gammaroidea mitogenomes in that dataset were incomplete, for comparative analyses, we retrieved all 15 available mitogenomes belonging to the infraorder Senticaudata (class Malacostraca: order Amphipoda) available in the RefSeq database (electronic supplementary material, Additional file S1; alignments available in the electronic supplementary material, Additional file S4). PhyloSuite was used to

batch-download all selected mitogenomes from the GenBank, extract genomic features, translate genes into amino acid sequences, and semi-automatically re-annotate ambiguously annotated tRNA genes with the help of the ARWEN output. The GenBank taxonomy was automatically replaced with the WoRMS database taxonomy using PhyloSuite, as the latter tends to be more up to date [67].

To test the stability of topology, we conducted phylogenetic analyses using different methods and datasets. Datasets used were: amino acid sequences of all 13 concatenated PCGs (PCG_AAs), nucleotides of concatenated 13 PCGs (PCG_NUC), and nucleotides of all mitochondrial genes (PCGRT: PCGs + rRNAs + tRNAs). Genes were aligned in batches with MAFFT [68], using '–auto' strategy, and 'codon' alignment mode for nucleotides and 'normal' mode for amino acid sequences of PCGs; whereas all RNA genes were aligned using Q-INS-i algorithm, which takes secondary structure information into account. PartitionFinder [69] was used to infer the best partitioning strategy and select best-fit evolutionary models for each partition using the greedy algorithm and AICC criterion (electronic supplementary material, Additional file S4). To infer the phylogeny, we used two standard methods, both of which assume compositional homogeneity of data (herein referred to as 'homogeneous' models), maximum-likelihood (ML) conducted using IQ-TREE [70] and Bayesian inference (BI) conducted using MrBayes 3.2.6 [71]. ML analysis was conducted with 100 standard bootstraps, and BI analysis was conducted with default settings, $2 \times 10^6$ metropolis-coupled MCMC generations (analysis was stopped as the average standard deviation of split frequencies was sufficiently low: 0.0024). All these analyses were conducted with the help of PhyloSuite. As compositional heterogeneity can compromise phylogenetic analyses, phylogeny was also inferred using the CAT-GTR site mixture model implemented in PhyloBayes-MPI 1.7a, which allows for site-specific rates of mutation (herein referred to as 'heterogeneous' model) [72]. This analysis was conducted on beta version of the Cipres server (https://cushion3.sdsc.edu/portal2/tools.action) [73], with default parameters (burnin = 500, invariable sites automatically removed from the alignment, two MCMC chains), -dgam set to 8, and the analysis was automatically stopped when the conditions considered to indicate a good run were reached: maxdiff less than 0.1 and minimum effective size greater than 300 (PhyloBayes manual). Phylograms and gene orders were visualized in iTOL [74] using files generated by PhyloSuite.

## 5.5. Selection pressure analyses

We used nucleotides of 13 PCGs (aligned using the codon mode; electronic supplementary material, Additional file S4) and PhyloBayes topology (figure 1) as the guidance tree. The numbers of segregating sites and mutations were estimated using DnaSP [75]. We tested a number of different evolutionary hypotheses using tools available from the Datamonkey webserver [76]: aBSREL to test if positive selection has occurred on a proportion of branches [38], BUSTED to test for a gene-wide (not site-specific) positive selection [39], RELAX to detect relaxed selection [40], and FEL to infer non-synonymous (dN) and synonymous (dS) substitution rates on a per-site basis [37].

Animal ethics. This study was approved by the Animal Care and Use Committee of the Centre for Applied Aquatic Genomics at Chinese Academy of Fishery Sciences.

Permission to carry out fieldwork. As sampling was conducted on unprotected invertebrates and on public land, fieldwork permits were not required.

Data accessibility. The two new mitogenomes are available from the GenBank repository under accession numbers MK354235 (*Gammarus lacustris*) and MK354236 (*Gammarus pisinnus*), and all other data generated or analysed during this study are included in this published article and its electronic supplementary material, information files.

Authors' contributions. S.S. and X.G. participated in the conceptualization and design of the study, collection of samples, laboratory work, data analysis and drafting the manuscript; Y.W. participated in the collection of samples, laboratory work, data analysis and critically revised the manuscript for important intellectual content; I.J. participated in the design of the study, data analysis, interpretation and visualization and drafting the manuscript; J.Z., S.M., K.A.A.G. and F.A.M. participated in the molecular laboratory work, data analysis and visualization, and critically revised the manuscript for important intellectual content; H.F. conceived and coordinated the study, and critically revised the manuscript for important intellectual content. All authors approved the final version and agree to be accountable for all aspects of the work.

Competing interests. We declare we have no competing interests.

Funding. This work was supported by National Natural Science Foundation of China (no. 31402280); the New Varieties Creation Major Project in Jiangsu Province (PZCZ201745); the Science & Technology Supporting Program of Jiangsu Province (BE2016308); the China Agriculture Research System-48 (CARS-48); and the Deanship of Scientific Research at King Saud University (1440-0138).

Acknowledgements. The authors express their sincere appreciation to the Deanship of Scientific Research at King Saud University for its funding of this research through the Research Group Project no. 1440-0138.

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
