## [Reviewer comments · Royal Society Open Science]

Review History

RSOS-190669.R0 (Original submission)

Review form: Reviewer 1

Is the manuscript scientifically sound in its present form?

No

Are the interpretations and conclusions justified by the results?

No

Is the language acceptable?

Yes

Is it clear how to access all supporting data?

Yes

Do you have any ethical concerns with this paper?

No

Have you any concerns about statistical analyses in this paper?

No

Recommendation?

Major revision is needed (please make suggestions in comments)

Comments to the Author(s)

Review for Manuscript RSOS-190669

"Disentangling the interplay of positive and negative selection forces that shaped mitochondrial genomes of *Gammarus pisinnus* and *Gammarus lacustris*."

Submitted by Sun et al.

The authors provide mitogenomic sequences for the high-altitude amphipod species *Gammarus pisinnus* and the low-altitude *Gammarus lacustris*.

They analyze both sequences in a phylogenetic context, along with other relatives at the level of family, in order to assess the occurrence, type and extent of selection driving the evolution of mitochondrial genomes in these species. To do so, the authors use the algorithms available in the Datamonykey webserver on a set of different data partitions. As outcome, the authors report purifying selection as the predominant force shaping the studied mitogenomes, as well as some particular taxa or partitions for which the occurrence of episodes of positive selection seem to have occurred.

Although the data are suitable to test the hypothesis of adaptation to hypoxic conditions in high-altitude environments, the methods chosen are, in my view, not the best suited. Moreover, the manuscript still needs some major revisions in the used methods that may lead to different results and discussion.

I detail below some concerns and make some suggestions for improvement.

A review of some terms used all along the manuscript would be highly valuable:

1- The terms 'positive' and 'diversifying' to describe the nature of selection are incorrectly used interchangeably. In the particular case tackled by this manuscript, the authors are rather looking for signatures of directional selection that would have favored the colonization of hypoxic, high-altitude environments, rather than signatures of disruptive or diversifying selection.

2- In page 9, lines 151-153: "Segregating sites", and "mutations" are not the best terms to describe what the authors intend to. My guess is that by 'segregating sites', the authors rather mean 'variable sites' ; and by 'mutations' they rather mean 'differences'.

Methods

It is important, and this aspect is missing in both the methods and the discussion, that the authors account for the strong variation in base composition reported across the 15 gammaroids included in the dataset. First, it violates the assumption of homogeneity on which rely the probabilistic substitution models used here to infer the phylogeny. Likewise, this is now well known to mislead selection tests, so it becomes important here to control for the effect of this variation on selection inference.

I would thus suggest that the authors include analyses using non-homogeneous models for both phylogeny inference and detection of selection to control for base composition. Mixture models might also be a good solution to overcome this constraint.

It is also unclear why the authors excluded the ribosomal and transfer RNAs from phylogenetic inference. These partitions are, due to their highly conserved sequences, usually well fitted to resolve deep branchings in the tree.

Also, as filtering was performed on alignments, more information is needed about the number of sites, fraction of missing data, etc, for each dataset used in the analyses.

It would also be interesting, as a potential additional outcome of the manuscript, to infer the phylogeny including all available taxa, specially in view of the apparent polyphyletic nature of Gammaroidea, described in the manuscript. Then, it would be preferable to have in Figure 1, the topology used in selection analyses, i. e. the one with the 15 Gammaroidea species + 1 outgroup, currently provided as Supp Fig 4 as it is the one underlying all the performed analyses. I suggest thus exchanging the placement of these two phylogenies.

As the authors acknowledge, detecting positive selection in mitogenomes is not straightforward, specially in view of the pervasive purifying selection regime on which they evolve. Moreover, statistical power is limited due to the reduced number of (linked) sites and, if partitioned, gene size and evolutionary rates are heterogeneous, making direct comparison difficult. These confounding factors are further complicated by the heterogeneity on strand-location, which creates asymmetric patterns of nucleotide composition evolution and in this case, by base composition. Altogether, these factors can sometimes mislead to inferring positive selection.

Regarding the methods used to assess the occurrence of selection, I could not avoid the feeling that, rather than carefully choosing the method to address a working hypothesis, the authors just went on with the whole package of algorithms available in the HyPhy server. I would suggest a refinement of the working hypothesis (or hypotheses) and a more critical choice of the methods to test them.

Results

Some of the reported start codons seem to be very uncommon in invertebrates (e.g. TTG). It might need to be further verified on electropherograms. Are these sites ambiguous? If dubious, these sites should be excluded from analyses as they can mislead selection tests.

I tried to access the sequences by using the provided accession numbers but it was not possible. Please verify if these numbers are still of actuality.

Review form: Reviewer 2 (Dmitry Yurievich Sherbakov)

Is the manuscript scientifically sound in its present form?

Yes

Are the interpretations and conclusions justified by the results?

No

Is the language acceptable?

Yes

Is it clear how to access all supporting data?

Yes

Do you have any ethical concerns with this paper?

No

Have you any concerns about statistical analyses in this paper?

Yes

Recommendation?

Major revision is needed (please make suggestions in comments)

Comments to the Author(s)

The manuscript addresses very interesting problem of animal adaptation to the extremal environments and possible genetic consequences/mechanisms of transformations involved. I have three major concerns about the manuscript:

- 1) Phylogenetic analysis of mitochondrial genomes based on the comparison of amino acid sequences of the 13 protein-coding genes taken as separate entities with different seems to be responsible for the unusual position of *G.fasciatus* due to small amount of variation per gene and too distant outgroups distorting substitution matrix. If all genes are pulled together and distant outgroups removed, all Micrurpodidae appear to be monophyletic but the set of position under selective pressure changes according to my analysis. Therefore I suggest that the 2 approaches (single model of substitutions frequencies (A) vs separate genes (B)) are more thoroughly justified based on the common statistic criteria (BIC, AIC...). If separate genes sequences are then used with topological constraint of the phylogeny inferred according to (A), the pattern of selective forces becomes different;
- 2) Since the author believe that hypoxia appears as the main environmental challenge to the hydrobionts adaptation to high altitude, it would be interesting to see the data on the oxygen concentrations in the waterbodies compared;
- 3) Using automated annotation of tRNA seems to be too simplistic since frequent tRNA remolding had been reported (Romanova et al, 2016)

Decision letter (RSOS-190669.R0)

05-Aug-2019

Dear Dr Jakovlic,

The editors assigned to your paper ("Disentangling the interplay of positive and negative selection forces that shaped mitochondrial genomes of *Gammarus pisinnus* and *Gammarus lacustris*") have now received comments from reviewers.

Both reviewers are positive about the publication of your paper, but both raise a number of very substantive points about the manuscript which will require careful attention and consideration. A number of issues are raised about the methodology and conclusions. In addition, it will be important to address the comment about availability of data. Overall, the manuscript will require major revision.

We would like you to revise your paper in accordance with the referee suggestions which can be found below (not including confidential reports to the Editor). Please note this decision does not guarantee eventual acceptance.

Please submit a copy of your revised paper before 28-Aug-2019. Please note that the revision deadline will expire at 00.00am on this date. If we do not hear from you within this time then it will be assumed that the paper has been withdrawn. In exceptional circumstances, extensions

may be possible if agreed with the Editorial Office in advance. We do not allow multiple rounds of revision so we urge you to make every effort to fully address all of the comments at this stage. If deemed necessary by the Editors, your manuscript will be sent back to one or more of the original reviewers for assessment. If the original reviewers are not available, we may invite new reviewers.

- Data accessibility

It is a condition of publication that all supporting data are made available either as supplementary information or preferably in a suitable permanent repository. The data accessibility section should state where the article's supporting data can be accessed, and all data should now be available; i.e. GenBank accessions must be live, and all data available for reviewers. This section should also include details, where possible of where to access other relevant research materials such as statistical tools, protocols, software etc can be accessed. If the data have been deposited in an external repository this section should list the database, accession number and link to the DOI for all data from the article that have been made publicly available. Data sets that have been deposited in an external repository and have a DOI should also be appropriately cited in the manuscript and included in the reference list.

If you wish to submit your supporting data or code to Dryad (<http://datadryad.org/>), or modify your current submission to dryad, please use the following link:
<http://datadryad.org/submit?journalID=RSOS&manu=RSOS-190669>

- Competing interests

- Authors' contributions

- Acknowledgements

- Funding statement

on behalf of Dr Steve Brown (Associate Editor) and Steve Brown (Subject Editor)
openscience@royalsociety.org

Reviewers' Comments to Author:

Reviewer: 1

Comments to the Author(s)

Review for Manuscript RSOS-190669

"Disentangling the interplay of positive and negative selection forces that shaped mitochondrial genomes of *Gammarus pisinnus* and *Gammarus lacustris*."

Submitted by Sun et al.

The authors provide mitogenomic sequences for the high-altitude amphipod species *Gammarus pisinnus* and the low-altitude *Gammarus lacustris*.

They analyze both sequences in a phylogenetic context, along with other relatives at the level of family, in order to assess the occurrence, type and extent of selection driving the evolution of mitochondrial genomes in these species. To do so, the authors use the algorithms available in the Datamonykey webserver on a set of different data partitions. As outcome, the authors report purifying selection as the predominant force shaping the studied mitogenomes, as well as some particular taxa or partitions for which the occurrence of episodes of positive selection seem to have occurred.

Although the data are suitable to test the hypothesis of adaptation to hypoxic conditions in high-

altitude environments, the methods chosen are, in my view, not the best suited. Moreover, the manuscript still needs some major revisions in the used methods that may lead to different results and discussion.

I detail below some concerns and make some suggestions for improvement.

A review of some terms used all along the manuscript would be highly valuable:

1- The terms 'positive' and 'diversifying' to describe the nature of selection are incorrectly used interchangeably. In the particular case tackled by this manuscript, the authors are rather looking for signatures of directional selection that would have favored the colonization of hypoxic, high-altitude environments, rather than signatures of disruptive or diversifying selection.

2- In page 9, lines 151-153: "Segregating sites", and "mutations" are not the best terms to describe what the authors intend to. My guess is that by 'segregating sites', the authors rather mean 'variable sites' ; and by 'mutations' they rather mean 'differences'.

Methods

It is important, and this aspect is missing in both the methods and the discussion, that the authors account for the strong variation in base composition reported across the 15 gammaroids included in the dataset. First, it violates the assumption of homogeneity on which rely the probabilistic substitution models used here to infer the phylogeny. Likewise, this is now well known to mislead selection tests, so it becomes important here to control for the effect of this variation on selection inference.

I would thus suggest that the authors include analyses using non-homogeneous models for both phylogeny inference and detection of selection to control for base composition. Mixture models might also be a good solution to overcome this constraint.

It is also unclear why the authors excluded the ribosomal and transfer RNAs from phylogenetic inference. This partitions are, due to their highly conserved sequences, usually well fitted to resolve deep branchings in the tree.

Also, as filtering was performed on alignments, more information is needed about the number of sites, fraction of missing data, etc, for each dataset used in the analyses.

It would also be interesting, as a potential additional outcome of the manuscript, to infer the phylogeny including all available taxa, specially in view of the apparent polyphyletic nature of Gammaroidea, described in the manuscript. Then, it would be preferable to have in Figure 1, the topology used in selection analyses, i. e. the one with the 15 Gammaroidea species + 1 outgroup, currently provided as Supp Fig 4 as it is the one underlying all the performed analyses. I suggest thus exchanging the placement of these two phylogenies.

As the authors acknowledge, detecting positive selection in mitogenomes is not straightforward, specially in view of the pervasive purifying selection regime on which they evolve. Moreover, statistical power is limited due to the reduced number of (linked) sites and, if partitioned, gene size and evolutionary rates are heterogeneous, making direct comparison difficult. This confounding factors are further complicated by the heterogeneity on strand-location, which creates asymmetric patterns of nucleotide composition evolution and in this case, by base composition. Altogether, these factors can sometimes mislead to inferring positive selection.

Regarding the methods used to assess the occurrence of selection, I could not avoid the feeling that, rather than carefully choosing the method to address a working hypothesis, the authors just went on with the whole package of algorithms available in the HyPhy server. I would suggest a refinement of the working hypothesis (or hypotheses) and a more critical choice of the methods to test them.

Results

Some of the reported start codons seem to be very uncommon in invertebrates (e.g. TTG). It might need to be further verified on electropherograms. Are these sites ambiguous? If dubious, these sites should be excluded from analyses as they can mislead selection tests.

I tried to access the sequences by using the provided accession numbers but it was not possible. Please verify if these numbers are still of actuality.

Reviewer: 2

Comments to the Author(s)

The manuscript addresses very interesting problem of animal adaptation to the extremal environments and possible genetic consequences/mechanisms of transformations involved. I have three major concerns about the manuscript:

- 1) Phylogenetic analysis of mitochondrial genomes based on the comparison of amino acid sequences of the 13 protein-coding genes taken as separate entities with different seems to be responsible for the unusual position of *G.fasciatus* due to small amount of variation per gene and too distant outgroups distorting substitution matrix. If all genes are pulled together and distant outgroups removed, all Micruropodidae appear to be monophyletic but the set of position under selective pressure changes according to my analysis. Therefore I suggest that the 2 approaches (single model of substitutions frequencies (A) vs separate genes (B)) are more thoroughly justified based on the common statistic criteria (BIC, AIC...). If separate genes sequences are then used with topological constraint of the phylogeny inferred according to (A), the pattern of selective forces becomes different;
- 2) Since the author believe that hypoxia appears as the main environmental challenge to the hydrobionts adaptation to high altitude, it would be interesting to see the data on the oxygen concentrations in the waterbodies compared;
- 3) Using automated annotation of tRNA seems to be too simplistic since frequent tRNA remodeling had been reported (Romanova et al, 2016)

Author's Response to Decision Letter for (RSOS-190669.R0)

See Appendix A.

RSOS-190669.R1 (Revision)

Review form: Reviewer 2 (Dmitry Yurievich Sherbakov)

Is the manuscript scientifically sound in its present form?

Yes

Are the interpretations and conclusions justified by the results?

Yes

Is the language acceptable?

Yes

Do you have any ethical concerns with this paper?

No

Have you any concerns about statistical analyses in this paper?

No

Recommendation?

Accept as is

Comments to the Author(s)

Interesting and useful paper.

Decision letter (RSOS-190669.R1)

29-Nov-2019

Dear Dr Jakovlic,

It is a pleasure to accept your manuscript entitled "Disentangling the interplay of positive and negative selection forces that shaped mitochondrial genomes of *Gammarus pisinnus* and *Gammarus lacustris*" in its current form for publication in Royal Society Open Science. The comments of the reviewer(s) who reviewed your manuscript are included at the foot of this letter.

Kind regards,
Lianne Parkhouse
Editorial Coordinator

on behalf of Professor Steve Brown (Subject Editor)
openscience@royalsociety.org

Reviewer comments to Author:

Reviewer: 2
Comments to the Author(s)

Interesting and useful paper.

Appendix A

Response to Referees

Reviewer: 1

Comments to the Author(s)

Review for Manuscript RSOS-190669

"Disentangling the interplay of positive and negative selection forces that shaped mitochondrial genomes of *Gammarus pisinnus* and *Gammarus lacustris*."

Submitted by Sun et al.

The authors provide mitogenomic sequences for the high-altitude amphipod species *Gammarus pisinnus* and the low-altitude *Gammarus lacustris*.

They analyze both sequences in a phylogenetic context, along with other relatives at the level of family, in order to assess the occurrence, type and extent of selection driving the evolution of mitochondrial genomes in these species. To do so, the authors use the algorithms available in the Datamonykey webserver on a set of different data partitions. As outcome, the authors report purifying selection as the predominant force shaping the studied mitogenomes, as well as some particular taxa or partitions for which the occurrence of episodes of positive selection seem to have occurred.

Although the data are suitable to test the hypothesis of adaptation to hypoxic conditions in high-altitude environments, the methods chosen are, in my view, not the best suited. Moreover, the manuscript still needs some major revisions in the used methods that may

lead to different results and discussion.

I detail below some concerns and make some suggestions for improvement.

A review of some terms used all along the manuscript would be highly valuable:

1- The terms 'positive' and 'diversifying' to describe the nature of selection are incorrectly used interchangeably. In the particular case tackled by this manuscript, the authors are rather looking for signatures of directional selection that would have favored the colonization of hypoxic, high-altitude environments, rather than signatures of disruptive or diversifying selection.

R: We were aware of this problem while writing the paper, so we opted to follow the terminology used in the publications associated with these analyses. For example, ABSREL (Smith et al. 2015) is "An adaptive branch-site REL test for episodic diversification". Then the authors say that "aBSREL (Adaptive branch-site random effects likelihood) uses an adaptive random effects branch-site model framework to test whether each branch has evolved under positive selection, using a procedure which infers an optimal number of rate categories per branch." And then for GL selected as the test branch, we get result that says "aBSREL found evidence of episodic diversifying selection on 1 out of 25 branches in your phylogeny." The common terms used for the three (provisional) types of evolution are 'directional', 'disruptive' and 'stabilising'. From this and the associated publications, we infer that aBSREL (and other used tools) tests for directional evolution, and not for disruptive

evolution, and uses “diversifying selection” as a synonym for directional evolution. As there seems to be no final agreement on the terminology, we opted to follow the one used by the authors of these tools in the corrected paper as well.

2- In page 9, lines 151-153: “Segregating sites”, and “mutations” are not the best terms to describe what the authors intend to. My guess is that by ‘segregating sites’, the authors rather mean ‘variable sites’ ; and by ‘mutations’ they rather mean ‘differences’.

R: In a similar fashion, we directly lifted the terms from the DnaSP output (<http://www.ub.edu/dnasp/DnaSPHelp.pdf>). In our experience, the ‘segregating sites’ term, defined as “The nucleotide sites that are polymorphic within a set of sequences”, is fairly commonly used in this context. We added ‘variable’ in brackets to clarify its meaning. We feel that the term ‘differences’ would be less specific than mutations in this context, so instead of replacing it, we added both terms to the M&M section, so that users can check them in the DnaSP documentation if they find them confusing: “The numbers of segregating sites and mutations were estimated using DnaSP [76].”

Methods

It is important, and this aspect is missing in both the methods and the discussion, that the authors account for the strong variation in base composition reported across the 15 gammaroids included in the dataset. First, it violates the assumption of homogeneity on

which rely the probabilistic substitution models used here to infer the phylogeny. Likewise, this is now well known to mislead selection tests, so it becomes important here to control for the effect of this variation on selection inference.

I would thus suggest that the authors include analyses using non-homogeneous models for both phylogeny inference and detection of selection to control for base composition. Mixture models might also be a good solution to overcome this constraint.

It is also unclear why the authors excluded the ribosomal and transfer RNAs from phylogenetic inference. These partitions are, due to their highly conserved sequences, usually well fitted to resolve deep branchings in the tree.

R: As inferring the correct phylogeny of Gammaroidea was not the main objective of our study, originally we merely conducted the standard (ML and BI) analyses. However, following your objections, and to assess the impact of compositional heterogeneity on our results, we conducted additional analyses using the 'heterogeneous' (please note that here and in the manuscript we use this term merely as a shortcut for 'model that accounts for compositional heterogeneity of data') CAT-GTR model implemented in PhyloBayes and added the RNA genes to the dataset. While producing several unresolved polyphyletic nodes, this analysis did manage to resolve Gammaridae as monophyletic, so we used this tree in the final paper, and re-conducted the evolutionary tests using this topology, to assess whether it affects the results. The results were mostly congruent, but with minor differences in p-values. Regardless, we thoroughly re-thought the entire Results and Discussion sections, and carefully revised them wherever we felt it was needed.

Also, as filtering was performed on alignments, more information is needed about the number of sites, fraction of missing data, etc, for each dataset used in the analyses.

R: As the impacts of alignment filtering appear to be dubious (Tan et al. 2015), and as the dataset used for the Datamonkey analyses was not filtered (in order to preserve the codon alignment), we decided not to use it at all in the repeated analyses.

It would also be interesting, as a potential additional outcome of the manuscript, to infer the phylogeny including all available taxa, specially in view of the apparent polyphyletic nature of Gammaroidea, described in the manuscript. Then, it would be preferable to have in Figure 1, the topology used in selection analyses, i. e. the one with the 15 Gammaroidea species + 1 outgroup, currently provided as Supp Fig 4 as it is the one underlying all the performed analyses. I suggest thus exchanging the placement of these two phylogenies.

R: As Gammaridae were the only paraphyletic taxon in the analysis of the larger Senticaudata dataset, and as the dataset that we used already comprises all available Gammaridae mitogenomes, we felt that having 'resolved' this issue (albeit, it should be noted that we cannot be certain that monophyletic gammarids are the correct topology), we felt that adding the analyses on the larger dataset would just produce noise in our paper. For the corrected paper, we used the Gammaroidea dataset for the new Figure 1 (monophyletic *Gammarus*), and shown results of other analyses as the Additional file 2.

As the authors acknowledge, detecting positive selection in mitogenomes is not straightforward, specially in view of the pervasive purifying selection regime on which they evolve. Moreover, statistical power is limited due to the reduced number of (linked) sites and, if partitioned, gene size and evolutionary rates are heterogeneous, making direct comparison difficult. This confounding factors are further complicated by the heterogeneity on strand-location, which creates asymmetric patterns of nucleotide composition evolution and in this case, by base composition. Altogether, these factors can sometimes mislead to inferring positive selection.

Regarding the methods used to assess the occurrence of selection, I could not avoid the feeling that, rather than carefully choosing the method to address a working hypothesis, the authors just went on with the whole package of algorithms available in the HyPhy server. I would suggest a refinement of the working hypothesis (or hypotheses) and a more critical choice of the methods to test them.

R: The difficulty in inferring positive selection with confidence is the exact reason why we used all of the tools available from Datamonkey server that were in any way relevant to our objectives. We feel that having results corroborated by several different algorithms is much more reliable than relying on a single algorithm. For the edited manuscript, we removed a part of the FEL analyses (individual genes), and only conducted them on genes for which we found indications that they may be evolving under unique selection pressures.

Results

Some of the reported start codons seem to be very uncommon in invertebrates (e.g. TTG). It might need to be further verified on electropherograms. Are these sites ambiguous? If dubious, these sites should be excluded from analyses as they can mislead selection tests.

R: That is not what was implied by our sentence. All these codons have been reported in gammaroids (please check comparative tables provided in the Additional file 1, worksheet B). Our sentence implied that they were novel for those specific genes in the dataset, but not for the dataset as a whole. Specifically, TTG is very common among the Gammaroidea, and other codons are standard as well. A possible exception is ATC, which still appears in approximately half of the available species, so we don't see any major reasons for suspicion here. Generally, we inspect electropherograms and repeat experiments whenever we encounter ambiguous results. We can submit them as supplementary data if required. As this appears to have caused confusion, and as there are no novel findings worthy of additional discussion here, we removed that sentence.

I tried to access the sequences by using the provided accession numbers but it was not possible. Please verify if these numbers are still of actuality.

R: It appears that we forgot to timely notify the GenBank staff to release the sequences. We apologise for this omission. Both sequences were automatically released on August 5th.

Thank you for investing time and expertise into reviewing our manuscript, your comments

helped us to improve it significantly.

Reviewer: 2

Comments to the Author(s)

The manuscript addresses very interesting problem of animal adaptation to the extremal environments and possible genetic consequences/mechanisms of transformations involved.

I have three major concerns about the manuscript:

1) Phylogenetic analysis of mitochondrial genomes based on the comparison of amino acid sequences of the 13 protein-coding genes taken as separate entities with different seems to be responsible for the unusual position of *G.fasciatus* due to small amount of variation per gene and too distant outgroups distorting substitution matrix. If all genes are pulled together and distant outgroups removed, all Micruropodidae appear to be monophyletic but the set of position under selective pressure changes according to my analysis. Therefore I suggest that the 2 approaches (single model of substitutions frequencies (A) vs separate genes (B)) are more thoroughly justified based on the common statistic criteria (BIC, AIC...). If separate genes sequences are then used with topological constraint of the phylogeny inferred according to (A), the pattern of selective forces becomes different;

R: In the corrected manuscript, we also added a phylogenetic analysis conducted using the CAT-GTR model, which allows for site-specific rates of mutation, which is considered to be a more realistic model of amino acid evolution, especially for large multi-gene alignments (Maddock et al. 2016). We also removed the distant outgroups. Indeed, we did obtain

monophyletic Gammarus, albeit with some unresolved polytomic nodes. We changed the Figure 1, results and discussion sections accordingly. We also re-conducted the selective analyses using the new topology. Although the results were mostly congruent, p-values exhibited minor differences. We thoroughly re-thought the entire Results and Discussion sections, and carefully revised them wherever we felt it was needed.

2) Since the author believe that hypoxia appears as the main environmental challenge to the hydrobionts adaptation to high altitude, it would be interesting to see the data on the oxygen concentrations in the waterbodies compared;

R: As we don't have data for this, and as high altitudes are not marked only by hypoxia, but also by low temperatures, high UV irradiation, etc., we replaced 'hypoxia' with 'adaptation to life at high altitudes'. As mitochondria are the powerhouse of the cell, numerous metabolic adaptations may be reflected upon their genomes, so we feel that this is a wiser approach to setting the objective of this study.

3) Using automated annotation of tRNA seems to be too simplistic since frequent tRNA remolding had been reported (Romanova et al, 2016)

R: We additionally inspected all tRNA alignments, and found no indication of tRNA remolding in the two new species. We added the following sentence to the Results section: "Although tRNA remolding has been reported in this group of crustaceans before [1], we

could not find evidence for such evolutionary events in the two newly sequenced species.”

Thank you for investing time and expertise into reviewing our manuscript, your comments helped us to improve it significantly.

References

Maddock, Simon T., Andrew G. Briscoe, Mark Wilkinson, Andrea Waeschenbach, Diego San Mauro, Julia J. Day, D. Tim J. Littlewood, Peter G. Foster, Ronald A. Nussbaum, and David J. Gower. 2016. “Next-Generation Mitogenomics: A Comparison of Approaches Applied to Caecilian Amphibian Phylogeny.” *PLoS ONE* 11 (6): e0156757. <https://doi.org/10.1371/journal.pone.0156757>.

Smith, Martin D., Joel O. Wertheim, Steven Weaver, Ben Murrell, Konrad Scheffler, and Sergei L. Kosakovsky Pond. 2015. “Less Is More: An Adaptive Branch-Site Random Effects Model for Efficient Detection of Episodic Diversifying Selection.” *Molecular Biology and Evolution* 32 (5): 1342–53. <https://doi.org/10.1093/molbev/msv022>.

Tan, Ge, Matthieu Muffato, Christian Ledergerber, Javier Herrero, Nick Goldman, Manuel Gil, and Christophe Dessimoz. 2015. “Current Methods for Automated Filtering of Multiple Sequence Alignments Frequently Worsen Single-Gene Phylogenetic Inference.” *Systematic Biology* 64 (5): 778–91. <https://doi.org/10.1093/sysbio/syv033>.

Manuscript with tracked changes

Disentangling the interplay of positive and negative selection forces that shaped mitochondrial genomes of *Gammarus pisinnus* and *Gammarus lacustris*

Formatted: English (United Kingdom)

Shengming Sun^{a,†}, Ying Wu^{a,†}, Xianping Ge^{a,b}, Ivan Jakovlić^c, Jian Zhu^{a,b}, Shahid Mahboob^{d,e}, Khalid Abdullah Al-Ghanim^d, Fahad Al-Misned^d, Hongtuo Fu^{a*}

^a Wuxi Fisheries College, Nanjing Agricultural University, Wuxi 214081, PR China

^b Agriculture Ministry Key Laboratory of Healthy Freshwater Aquaculture, Zhejiang Institute of Freshwater Fisheries, Huzhou 313001, China

^c Bio-Transduction Lab, Wuhan, China

^d Department of Zoology, College of Science, King Saud University, P.O. Box 2455, Riyadh-11451, Riyadh, Saudi Arabia

^e Department of Zoology, GC University, Faisalabad, Pakistan

*Corresponding author (H. Fu): E-mail: fuht@ffrc.cn; Tel./fax: +86 0510 85558835

Formatted: English (United Kingdom)

[†] These authors contributed equally to the work.

Running head: Evolution of two ~~*Gammarus*~~ *Gammarus* mitogenomes

Formatted: Font: 11 pt, English (United Kingdom)

Formatted: Font: Italic, English (United Kingdom)

Formatted: Font: 11 pt, Italic, English (United Kingdom)

Formatted: Font: Italic, English (United Kingdom)

Formatted: Font: 11 pt, English (United Kingdom)

Formatted: English (United Kingdom)

Abstract

We hypothesised that the mitogenome of *Gammarus lacustris* (GL), native to the Qinghai-Tibet Plateau, might exhibit genetic adaptations to the extreme environmental conditions associated with high altitudes (>3000 m). To test this, we also sequenced the mitogenome of *Gammarus pisinnus* (GP), whose native range is close to the Tibetan plateau, but at a much lower altitude (200-1500 m). The two mitogenomes exhibited conserved mitochondrial architecture, but low identity between genes (55% *atp8* to 76.1% *cox1*). Standard ('homogenous') phylogenetic models resolved Gammaridae as paraphyletic, but 'heterogeneous' CAT-GTR model as monophyletic. In indirect support for our working hypothesis, *Gammarus lacustris*, *pisinnus* and *fossarum* exhibit evidence of episodic diversifying selection within the studied Gammaroidea dataset. The mitogenome of GL generally evolves under a strong purifying selection, whereas GP evolves under directional (especially pronounced in *atp8*) and/or relaxed selection. This is surprising, as GP does not inhabit a unique ecological niche compared to other gammarids. We propose that this rapid evolution of the GP mitogenome may be a reflection of its relatively recent speciation and heightened nonadaptive (putatively metabolic rate-driven) mutational pressures. To test these hypotheses, we urge sequencing mitogenomes of remaining *Gammarus* species populating the same geographic range as GP.

~~We hypothesised that the mitogenome of the amphipod crustacean *Gammarus lacustris* (GL), native to the Qinghai-Tibet Plateau, might exhibit genetic adaptations to the extreme environmental conditions associated with life at high altitudes (>3000-4500 m), primarily to hypoxia. We also sequenced the mitogenome of *Gammarus pisinnus* (GP), whose native~~

Formatted: Line spacing: Double

range is relatively close to the Tibetan plateau, but at a much lower altitude (200–1500 m). The two mitogenomes exhibited low identity values between genes (55% *atp8* to 76.1% *cox1*) and gene families (*cox* 73.8% > *nad* 62.8% > *atp* 60.84%). Synonymous/nonsynonymous mutation analyses suggested intense purifying selection in the *cox* family, and somewhat relaxed selection in the *nad* family of GL. With the exception of *atp8*, the mitogenome of GP predominantly exhibited relaxed selection. Within the studied Gammaroidea dataset, we found evidence of episodic diversifying selection in three *Gammarus* species: *fossarum*, *pisinnus* and *lacustris*. Results provide a weak support for our working hypothesis, and indicate a much stronger positive directional selection evolution signal for GP than for GL. We propose that the signal of episodic positive selection in the mitogenome of GL has been swamped by 70 million years (uplift of the Tibetan plateau) of purifying selection, but not in GP, the speciation of which took place only 6.4 million years ago.

Keywords: mitogenome, Gammaridae, paraphyletic, monophyletic, high altitude adaptation, Tibet, atp8, nonadaptive mutational pressure OXPHOS, hypoxic environment

Background

The amphipod crustacean genus *Gammarus* Fabricius, 1775 (Senticaudata: Gammaroidea: Gammaridae) contains more than 200 described species [2], which makes it one of the most speciose crustacean genera [3]. These species are widely distributed throughout the Northern hemisphere in various marine, brackish, freshwater and subterranean aquatic habitats [4]. *Gammarus lacustris* (GL) is a freshwater species with a nearly circumboreal range [2], widely distributed in alpine lakes, and native to the Qinghai-Tibet Plateau (QTP) [4]. The QTP is the highest and largest plateau in the world, on average about 3000-4500 m in altitude [5]. As opposed to GL, *Gammarus pisinnus* (GP) inhabits rivers and lakes in a narrow geographical range comprising the plains south of the Taihang mountains, along the Weihe River in Shanxi Province [6], several hundred kilometers eastward from the eastern edge of the QTP, and at a much lower altitude of 200 to 1500 m [6]. The geological ages of these two habitats are also very different: the uplift of the Tibetan plateau began during the Upper Cretaceous, some 70 MYA (million-years-ago), whereas the uplift of the Taihang mountain range occurred during the Miocene (23 to 5 MYA) [4, 6]. The uplifting of the QTP dramatically changed the environmental conditions from an originally humid and warm climate to the currently dry and cold one [7]. Such dramatic ecological changes are expected to result either in the extinction of a species from the habitat or in major genetic and phenotypic adaptations to the new environment. Indeed, the QTP is typically inhabited by endemic species [8, 9] highly adapted to the extreme environmental conditions, such as hypoxia, low temperatures and strong ultraviolet radiation [4, 5, 10-15]. Although amphipod crustaceans are the predominant macroscopic aquatic invertebrate in the QTP [16],

Formatted: English (United Kingdom)

Formatted: English (United Kingdom)

Formatted: English (United Kingdom)

Formatted: English (United Kingdom)

Formatted: English (United Kingdom)

Formatted: English (United Kingdom)

Formatted: English (United Kingdom)

Formatted: English (United Kingdom)

Formatted: English (United Kingdom)

Formatted: English (United Kingdom)

Formatted: English (United Kingdom)

Formatted: English (United Kingdom)

Formatted: English (United Kingdom)

Formatted: English (United Kingdom)

Formatted: English (United Kingdom)

Formatted: English (United Kingdom)

Formatted: English (United Kingdom)

Formatted: English (United Kingdom)

Formatted: English (United Kingdom)

Formatted: English (United Kingdom)

Formatted: English (United Kingdom)

Formatted: English (United Kingdom)

Formatted: English (United Kingdom)

Formatted: English (United Kingdom)

Formatted: English (United Kingdom)

Formatted: English (United Kingdom)

Formatted: English (United Kingdom)

Formatted: English (United Kingdom)

Formatted: English (United Kingdom)

specificities of the high altitude adaption of crustaceans remain poorly understood. Furthermore, GL has been found at Tibetan localities exceeding elevations of 5000 m, which is currently the highest known altitudinal record for gammarids [9]-[8, 9]. This exceptional ability of GL to inhabit extremely high altitudes, matched only by some molluscs among the aquatic invertebrates [8, 9], is an indication that this species may be a very good model to investigate genetic adaptations to the high-altitude environment.

- Formatted: English (United Kingdom)
- Formatted: English (United Kingdom)
- Formatted: English (United Kingdom)
- Formatted: English (United Kingdom)
- Formatted: English (United Kingdom)
- Formatted: English (United Kingdom)

Owing to the abundance of mitochondria in cells, (mostly) maternal inheritance, absence of introns, small genomic size (in metazoans), and an increasingly large set of available orthologous sequences, mitochondrial genomes (mitogenomes) have become a popular tool in population genetics [17], phylogenetics [18-21], diagnostics [21-23], and evolution [24-27]. Although long-considered to evolve under the purifying selection, there is increasing evidence that mitogenomes can undergo episodes of positive-directional selection in response to shifts in physiological or environmental pressures [28]. The As a number of signs of genetic adaptation to hypoxia have been identified in Tibetan mammals [12, 13, 29-31] and fish [32], as m mitochondrial genome harbours genes involved in oxidative phosphorylation (OXPHOS), and thus plays a central role in the energy production in the organism [33]. Due to low temperatures and hypoxia, life at high altitudes is expected to generate unique evolutionary pressures on the energy management system of organisms [10, 11]. However, high-altitude adaptation can occur on different metabolic levels, not all of which include the mitochondrial metabolism [31, 34], so mitochondrial genome may reflect only a small fragment of these adaptive changes. Despite this limitation, and as a number of studies have detected positive-selection signals of directional selection in the

- Formatted: English (United Kingdom)
- Formatted: English (United Kingdom)
- Formatted: English (United Kingdom)
- Formatted: English (United Kingdom)
- Formatted: English (United Kingdom)
- Formatted: English (United Kingdom)
- Formatted: English (United Kingdom)
- Formatted: English (United Kingdom)
- Formatted: English (United Kingdom)
- Formatted: English (United Kingdom)
- Formatted: English (United Kingdom)
- Formatted: English (United Kingdom)
- Formatted: English (United Kingdom)
- Formatted: English (United Kingdom)
- Formatted: English (United Kingdom)
- Formatted: English (United Kingdom)
- Formatted: English (United Kingdom)
- Formatted: English (United Kingdom)

mitogenomes of animals living at high elevations [34-36], ~~including both terrestrial~~ [12, 13, 29-31] ~~and aquatic~~ [32] ~~Tibetan fauna~~. Therefore, ~~our working hwe hypothesis~~ ~~ed was~~ that the speciation of *G. lacustris* (but not that of *G. pisinnus*) ~~may have been~~ ~~was~~ accompanied by mitogenomic adaptations to ~~the extreme environment of the Tibetan plateau~~ ~~a hypoxic high-altitude environment~~. To test this hypothesis, we sequenced the complete mitochondrial genomes of *Gammarus lacustris* and *Gammarus pisinnus*, ~~and added them to a set of available gammarid mitogenomes to~~ ~~conduct~~ ~~ed~~ comparative mitogenomic and phylogenomic analyses ~~using a set of available gammarid mitogenomes~~, and ~~studied~~ ~~ied~~ evolutionary signals imprinted ~~upon the two gammarid~~ mitogenomes.

Formatted: English (United Kingdom)

Formatted: English (United Kingdom)

Formatted: English (United Kingdom)

Formatted: English (United Kingdom)

Formatted: English (United Kingdom)

Formatted: English (United Kingdom)

Results

Characteristics and architecture of the two mitogenomes

The length of complete mitochondrial genomes was: 15,907 bp in *G. pisinnus* (GP), 15,333 bp in *G. lacustris* (GL). Both possessed the standard 13 protein-coding genes, two rRNA genes (16S and 12S), 22 tRNA genes and a control region. Both mitogenomes exhibited an identical, highly conserved, gene order and strand distribution (Table 1). The genes of the two mitogenomes also exhibited very similar sizes, partially shared start/stop codons, and very similar intergenic regions (IGR) and overlaps. Size-wise, there were a few exceptions: *rrnS* gene was larger in GL than in GP (750 vs. 717 bp), whereas *nad1* was much shorter in GL (912 bp) than in GP (933 bp); GP had a much larger IGR between *trnI* and *trnC* (191 vs. 13 bp) and a much larger (putative) control region (1376 vs. 981 bp). Several start codons were found, ATG, TTG, ATA, ATT GTG and ATC, ~~w, and hereas~~ most genes used the standard TAA

stop codon (including an abbreviated form of it T--). An, with the exception was of *nad3* of both species and *nad4L* of GP, which used TAG. GP had the A+T content of 70% and GL of 64.3%. None of these The architectural features are novel or unique described above is also mostly standard for among the Gammaroidea and Senticaudata (Additional file 21), with the exception of some start/stop codons: ATC as the start codon for *cox1* (GL), TTG for *atp6* (GP) and ATT for *cox2* (GP), and TAG stop codon for the *nad4L* of GP and *nad3* of both species (Additional file 2).

Table 1. Organisation of the mitogenomes of *Gammarus lacustris* and *Gammarus pisinnus*.

[revised manuscript text omitted]

highest value of 76.1%. As regards gene families, *cox* family (*cox1-3*) exhibited the highest average value (73.8%), followed by the *nad* family (62.8%), and finally the *atp* family (60.84%). Although tRNA remodeling has been reported in this group of crustaceans before [1], we did not find evidence for such evolutionary events in the two newly sequenced species.

Formatted: English (United Kingdom)

Formatted: English (United Kingdom)

Formatted: English (United Kingdom)

Phylogeny and gene order

Gene order was perfectly conserved in the entire Gammaridae family, including the two newly sequenced species (Figure 1). Different datasets and algorithms produced notable topological instability. Apart from the CAT-GTR analysis of the PCGRT dataset (Figure 1), other analyses produced deeply paraphyletic family Gammaridae, mostly with three or even four Baikal lake families (Acanthogammaridae, Pallaseidae, Eulimnogammaridae + Micruropodidae) nested within the clade (Additional file 2). Two analyses, Maximum Likelihood (ML) and Bayesian Inference (BI) (alignment, model selection results and analysis settings are available in the Additional file 1, and species details in the Additional File 2), produced congruent topologies (Figure 1), with only one exception: ML resolved GP and *Gmelinoides fasciatus* as sister taxa (Additional file 3). Both analyses produced a deeply paraphyletic Gammaridae, with three Baikal lake families (Acanthogammaridae, Pallaseidae and Eulimnogammaridae) nested within the clade. In the BI tree, GP was the basal species in the Gammaroidea clade, whereas GL and *G. duebeni* were sister taxa. CAT-GTR model resolved Gammaridae as monophyletic, notwithstanding several nodes in the phylogram that exhibited unresolved polyphyletic relationships. Apart from Acanthogammaridae, which were rendered paraphyletic by the nested Pallaseidae (*Pallaseopsis kessleri*), family, the

Formatted: Font: Italic, English (United Kingdom)

Formatted: English (United Kingdom)

remaining Gammaroidea families (Micruropodidae and Eulimnogammaridae) were also monophyletic (taxonomic details available in Additional files 1 and 2).

With the exception of a *trnD* translocation in *Brachyropus grewingkii*, gene order was perfectly conserved in the entire Gammaroidea superfamily (Figure 1).

Formatted: English (United Kingdom)

Figure 1. Mitochondrial phylogenomics and gene orders of the superfamily Gammaroidea Senticaudata. Phylogram was inferred using nucleotide sequences of all mitochondrial genes (PCG+rRNA+tRNA) and CAT-GTR model implemented in PhyloBayes through a Bayesian analysis of amino acid sequences of all 13 protein coding genes. *Limulus polyphemus*, *Squilla mantis* and *Cymothoa indica* are *Metacrangonyx remyi* is the outgroups (branches cropped).

Formatted: Font: Italic, English (United Kingdom)

Formatted: English (United Kingdom)

Scale bar corresponds to the estimated number of substitutions per site. Posterior probability support values <1.0 are shown next to nodes, and GenBank (RefSeq) accession numbers are available in the Additional file 1 next to species names. Gene orders are shown to the right, with the legend included in the figure.

Direction and magnitude of selective pressures

The rate of nonsynonymous (dN) to synonymous (dS) substitutions, $\omega = dN/dS$, is a commonly used indicator for measuring the direction and magnitude of selective pressure on genes, where $\omega < 1$ indicates purifying selection, $\omega = 1$ indicates neutral evolution, and $\omega > 1$ indicates directional (or positive) selection [37]. ~~To minimize the impact of mutational saturation, for these analyses we focused only on the 15 available mitogenomes belonging to the superfamily Gammaroidea, plus *Metacrangonyx remyi* [38] as outgroup (Additional file 2). To test whether different genes and gene families may be evolving under different selection pressures, we tested 17 different datasets: concatenated 13 PCGs, concatenated atp family (atp6 and 8), cox family (cox1-3) and nad family (nad1-6) genes, and all 13 individual genes separately (1+3+13=17). In the dataset comprised of 13 concatenated PCGs, the number of segregating (variable) sites —was 1283, total number of observed mutations was 2061, among which 1865 were synonymous, and 196 were nonsynonymous.~~

First we used aBSREL, which allows a global test for positive selection in the entire dataset [39]. Using the concatenated 13 PCGs dataset with all 15 species (apart from the outgroup *M. remyi*) selected as test branches, we found statistically significant evidence (p-value corrected for multiple testing ≤ 0.05) of episodic diversifying selection in two species: *Gammarus fossarum* ($p=0.0026$), and the newly sequenced GP ($p=0.0095$). The corrected p-value of GL was also very close to the significance threshold ($p=0.0658$; the uncorrected p-value was 0.05). For the global test, p-values at each branch must be corrected for multiple testing, which decreases the selection detection power of this

Formatted: English (United Kingdom)

Formatted: English (United Kingdom)

Formatted: English (United Kingdom)

Formatted: English (United Kingdom)

Formatted: English (United Kingdom)

Formatted: English (United Kingdom)

exploratory approach [39], so we then used the same tool to conduct independent tests for GL and GP (selected as the test branch), and the remaining species as reference branches. Results of these two tests were expected, within the context of the global test: evidence of episodic diversifying selection was identified in both species (p-values: GL=0.0052, GP=0.0007).

Formatted: English (United Kingdom)

Formatted: English (United Kingdom)

Using the BUSTED tool, which “provides a gene-wide (not site-specific) test for positive selection by asking whether a gene has experienced positive selection at at least one site on at least one branch” [40], we found evidence of gene-wide episodic diversifying selection in both species (independently selected as test branches): p=0.048 for GL, and p=0.001 for GP.

Formatted: English (United Kingdom)

Formatted: English (United Kingdom)

Relaxation of the efficiency or intensity of natural selection can result in a loss of function, but also drive evolutionary innovation [41], so we also explored the direction and magnitude of selection pressures on GL and GP using the RELAX so we tested the hypotheses of relaxed/intensified selection in the mitogenome of GL within a codon-based phylogenetic framework. First, we tested all 15 branches in the guidance tree (Additional file 4) for positive selection, with only *M. remyi* (outgroup) selected as the background. For this analysis we used aBSREL, which can test if positive selection has occurred on a proportion of branches [39]. Using the concatenated 13 PCGs dataset, we found evidence ($p \leq 0.05$) of episodic diversifying selection in three species: *Gammarus fossarum* (p=0.0005), and both newly sequenced mitogenomes, GP (p=0.0008) and GL (p=0.0027). However, when we broke down the dataset into different subsets, aBSREL did not find evidence of episodic diversifying selection in GL, but it did find evidence in the *atp8* of *G. pisinnus* (p=0.00181). It also found evidence in some other species: *nad* gene family in *G. fossarum* (p=0.0105), *atp6*

Formatted: English (United Kingdom)

Formatted: English (United Kingdom)

Formatted: English (United Kingdom)

Formatted: English (United Kingdom)

of *Gammarus duebeni* ($p=0.0001$), *atp8* of *L. vortex* ($p=0.0009$), *nad1* of *G. fossarum* ($p=0.0269$), and *nad3* of *A. victoria* ($p=0.0091$) and *G. fossarum* ($p=0.0006$). For the global test, p values at each branch must be corrected for multiple testing, which decreases the selection detection power of this exploratory approach [39]. Therefore, we then used the same tool to conduct independent tests for GL and GP (selected as the test branch independently), and the remaining species as reference branches. Results of these two tests were congruent with the global test: evidence of episodic diversifying selection was identified only in the 13PCG dataset in GL, and in 13PCG and *atp8* datasets in GP (Table 2).

Formatted: English (United Kingdom)

Formatted: English (United Kingdom)

We further explored the direction and magnitude of selection pressures on GL and GP (two tests, each with one species selected as the test branch) using two additional tools: BUSTED and RELAX. BUSTED tests whether a gene has experienced positive selection at at least one site on at least one branch [40]. With GL selected as the test branch, we did not find evidence for diversifying selection along the test branch in any of the datasets, but we found evidence in the 13PCG, *nad* family and *atp8* datasets, with GP selected as the test branch (Table 2). Following this, we applied RELAX algorithm, which asks whether the strength of natural selection has been relaxed or intensified along a specified set of test branches. W[41]. While this method is not suitable for explicitly testing for positive selection, this method is it is useful for identifying trends and/or shifts in the stringency of natural selection on a given locus [41]. A significant result of $k>1$ indicates that selection strength has been intensified along the test branches, and a significant result of $k<1$ indicates that selection strength has been relaxed along the test branches. Both analyses produced statistically significant results, but the tests indicate selection intensification in GL,

Formatted: None, Indent: First line: 1.75 pt, Space Before: 0 pt, No widow/orphan control, Don't keep with next, Don't keep lines together

Formatted: English (United Kingdom)

Formatted: English (United Kingdom)

Formatted: English (United Kingdom)

Formatted: English (United Kingdom)

Formatted: English (United Kingdom)

Formatted: English (United Kingdom)

and selection *relaxation* in GP (Table 2).

Formatted: Font: Italic

To further explore the selection signals using a per site based approach we relied on the FEL algorithm, which uses a maximum likelihood approach to infer dN and dS substitution rates on a per site basis for a given coding alignment and corresponding phylogeny [37]. In GL, the sites exhibiting predominantly negative (purifying) selection by far outnumbered the sites exhibiting predominantly positive (diversifying) selection (997 vs 17 respectively). In GP, the situation was inverted: sites exhibiting pervasive positive/diversifying selection by far outnumbered the sites exhibiting pervasive negative/purifying selection (619 vs 0 respectively; $p < 0.05$). Significance threshold was set at $p < 0.05$ for both analyses.

Formatted: English (United Kingdom)

Formatted: English (United Kingdom)

Formatted: English (United Kingdom)

To get a closer perspective, and test which other different genes may be evolving under unique different selection pressures, we divided the dataset into individual genes and conducted out of 17 tested datasets, another round of RELAX analyses. To test whether

Formatted: English (United Kingdom), Highlight

Formatted: English (United Kingdom)

different genes and gene families may be evolving under different selection pressures, we tested 17 different datasets: concatenated 13 PCGs, concatenated atp family (*atp6* and *8*), *cox* family (*cox1-3*) and *nad* family (*nad1-6*) genes, and on all 13 individual genes (Table 2) separately (1+3+13=17). We did not find statistically significant evidence of directional

Formatted: Font: Italic, English (United Kingdom)

Formatted: English (United Kingdom)

selection (aBSREL+BUSTED) in any of the GL genes, but p-values indicate that *cox3* may be undergoing higher levels of directional selection pressures than other genes. RELAX analyses

Formatted: Font: Italic, English (United Kingdom)

Formatted: English (United Kingdom)

Formatted: Font: Italic, English (United Kingdom)

Formatted: English (United Kingdom)

revealed that all GL genes except *nad4L* (significant) and *nad6* (non-significant) exhibited intensified selection (six significant: *atp6*, *cox1*, *cox3*, *cytb*, *nad3* and *nad5*). We found statistically significant evidence for selection in seven seven datasets in genes in GL, six all

Formatted: English (United Kingdom), Highlight

Formatted: English (United Kingdom)

of which exhibited intensified selection pressure (Table 2). Among the four concatenated datasets, only the nad family did not exhibit signs of intensified selection. This was further corroborated by tests on individual genes, where *atp6*, *cox1*, *cox3*, and *cytb*, *nad3* and *nad5*, whereas only *nad4L* exhibited a significant intensified relaxed selection pressure. The third *cox* family gene, *cox2*, was not an outlier, as it also exhibited a highly intensified selection, just above the significance threshold ($p=0.069$) (Table 2). In GP, *atp8* (aBSREL+BUSTED) and *nad3* (aBSREL) exhibited statistically significant evidence of directional selection. Most genes exhibited different levels of relaxed selection pressures (we found statistically significant evidence for selection in three genes (*cytb*, *nad4* and *nad6*), with only *cox1*, *nad4L* and *nad5* exhibiting (non-significantly) intensified selection pressures, all of which exhibited relaxed selection pressure (Table 2). –

To further explore the selection signals using a per-site-based approach in selected genes of interest (Table 3), we relied on the FEL algorithm, which uses a maximum-likelihood approach to infer dN and dS substitution rates on a per-site basis for a given coding alignment and corresponding phylogeny [37]. First we tested the entire dataset, and then genes for which other analyses (Table 2) indicated that they may be evolving under unique selection pressures (significance threshold was set at $p<0.05$ for all analyses). For the entire dataset (13PCG) GL, the sites exhibiting predominantly negative (purifying) selection by far outnumbered the sites exhibiting predominantly positive (diversifying) selection (897 vs 17 respectively) in GL, but in GP, the situation was inverted in GP, where sites exhibiting pervasive positive/diversifying selection by far outnumbered the sites exhibiting pervasive negative/purifying selection (619 vs 0 respectively; $p<0.05$). This was

Formatted: Font: Italic, English (United Kingdom)

Formatted: English (United Kingdom)

Formatted: Font: Italic, English (United Kingdom)

Formatted: English (United Kingdom)

Formatted: Font: Italic, English (United Kingdom)

Formatted: English (United Kingdom)

Formatted: English (United Kingdom)

Formatted: English (United Kingdom)

reflected in individual genes as well, where all three GP genes (*atp8*, *cox3* and *nad3*) exhibited only positive selection. In GL, *nad4L* and *nad6* also exhibited only positive selection, whereas *cox3* exhibited only negative selection. Significance threshold was set at $p < 0.05$ for both analyses.

Finally, to better examine the signal of episodic diversifying selection in *Gammarus fossarum* observed in the exploratory aBSREL test ($p = 0.0026$), we conducted a few additional analyses with this species selected as the test branch. aBSREL ($p = 0.0002$) and BUSTED found evidence ($p = 0.000$) of episodic diversifying selection. RELAX test for selection relaxation ($K = 0.84$) was also significant ($p = 0.001$, $LR = 11.52$), which was supported by the FEL analysis, where sites exhibiting pervasive positive/diversifying selection by far outnumbered the sites exhibiting pervasive negative/purifying selection (346/0 respectively; $p < 0.05$).

The individual nad family members exhibited a mix of intensified (1, 3 and 6) and relaxed (2, 4, 4L and 5) selection, but none of the p-values were significant; although *nad3* (intensified) and *nad4L* (relaxed) genes exhibited p-values very near the significance threshold (0.063 and 0.050 respectively). *Atp6* exhibited a significantly intensified selection, but *atp8* exhibited a relaxed selection (non-significantly). In GP, we found significant evidence for relaxed selection in 13PCG and nad family datasets, and intensified selection in *atp8*.

Table 2 Selection signals in the mitogenomes of *Gammarus lacustris* (GL) and *Gammarus pinnatus* (GP) inferred using BUSTED, aBSREL, and RELAX algorithms

- Formatted: Font: Not Bold, English (United Kingdom)
- Formatted: English (United Kingdom)
- Formatted: Font: Not Bold, English (United Kingdom)
- Formatted: English (United Kingdom)
- Formatted: Font: Not Bold, English (United Kingdom)
- Formatted: English (United Kingdom)
- Formatted: Font: Not Bold, English (United Kingdom)
- Formatted: English (United Kingdom)
- Formatted: Font: Not Bold, English (United Kingdom)
- Formatted: English (United Kingdom)

~~13PCG is the dataset comprising all 13 concatenated (non-partitioned) protein-coding genes.~~
~~Sites column shows the number of sites in the alignment. K column: a statistically significant~~
 ~~$K > 1$ indicates that selection strength has been intensified, and $K < 1$ that selection strength~~
~~has been relaxed. LR is likelihood ratio, p is p-value, and D indicates the direction of selection~~
~~pressure change: intensified (I) or relaxed (R), where * highlights a statistically significant~~
~~($p < 0.05$) result.~~

1 **Table 2. Selection signals in the mitogenomes of *Gammarus lacustris* (GL) and *Gammarus pisinnus* (GP) inferred using BUSTED, aBSREL, and RELAX**

**algorithms.**

13PCG is the dataset comprising all 13 concatenated (non-partitioned) protein-coding genes. Sites column shows the number of sites in the alignment. K
 column: a statistically significant K>1 indicates that selection strength has been intensified, and K<1 that selection strength has been relaxed. LR is likelihood
 ratio, p is p-value, and D indicates the direction of selection pressure change: intensified (I) or relaxed (R), where * highlights a statistically significant
 (p<0.05) result.

Data	Sites	Gammarus lacustris (GL)						Gammarus pisinnus (GP)					
		BUSTED		aBSREL	RELAX			BUSTED		aBSREL	RELAX		
		p	p	K	LR	p	D	p	p	K	LR	p	D
13PCG	3743	0.024*	0.005*	1.21	11.54	0.001	I*	0.001*	0.010*	0.63	56.39	0.000	R*
atp6	223	1.000	1.000	14.80	7.48	0.006	I*	0.487	1.000	0.68	1.51	0.219	R
atp8	58	1.000	1.000	1.02	0.02	0.897	I	0.000*	0.004*	0.13	1.14	0.286	R
cox1	527	1.000	1.000	1.87	12.31	0.000	I*	1.000	1.000	1.00	-0.38	1.000	I
cox2	231	1.000	1.000	1.60	2.76	0.097	I	1.000	1.000	1.00	0.00	1.000	R
cox3	262	0.345	0.087	2.24	8.05	0.005	I*	0.327	0.144	0.45	-6.96	1.000	R
cytb	380	1.000	0.168	2.37	6.63	0.010	I*	1.000	1.000	0.46	7.91	0.005	R*

Formatted: Left: 6 pi, Right: 6 pi, Top: 7.5 pi, Bottom: 7.5 pi, Width: 70.16 pi, Height: 49.61 pi

Formatted: English (United Kingdom)

Formatted: English (United Kingdom)

Formatted: English (United Kingdom)

Formatted: English (United Kingdom)

Formatted: English (United Kingdom)

Formatted: English (United Kingdom)

Formatted: English (United Kingdom)

Formatted: English (United Kingdom)

Formatted: English (United Kingdom)

Formatted: English (United Kingdom)

pad1	313	1.000	1.000	1.18	0.90	0.343	I	0.749	1.000	0.77	3.24	0.072	R
pad2	330	0.718	1.000	1.01	0.37	0.543	I	0.562	0.431	0.84	0.42	0.516	R
pad3	119	1.000	1.000	7.91	6.98	0.008	I*	0.049*	0.454	0.64	2.28	0.131	R
pad4	447	1.000	1.000	1.04	0.40	0.530	I	1.000	1.000	0.58	7.75	0.005	R*
pad4L	98	1.000	1.000	0.36	4.98	0.026	R*	1.000	1.000	2.05	2.61	0.106	I
pad5	579	0.741	0.385	1.38	4.23	0.040	I*	0.460	0.500	2.15	-45.82	1.000	I
pad6	176	0.766	0.500	0.95	-0.09	1.000	R	1.000	1.000	0.49	4.41	0.036	R*

Formatted: English (United Kingdom)

Formatted: English (United Kingdom)

Formatted: English (United Kingdom)

Formatted: English (United Kingdom)

Formatted: English (United Kingdom)

Formatted: English (United Kingdom)

Formatted: English (United Kingdom)

Formatted: English (United Kingdom)

**Table 3. Selection signals in the selected genes of mitogenomes of *Gammarus lacustris* (GL)**
 **and *Gammarus pisinnus* (GP) inferred using a per-site-based method – FEL. 13PCG is the**
 **dataset comprising all 13 concatenated (non-partitioned) protein-coding genes. Sites column**
 **shows the number of sites in the alignment. + indicates the number of sites evolving under**
 **significant (threshold set at $p < 0.05$) pervasive positive/diversifying selection, and - under**
 **pervasive negative/purifying selection.**

Data	Sites	GL		GP	
		+	-	+	-
13PCG	3743	17	897	619	0
atp8	58			18	0
cox3	262	0	78	37	0
nad3	119			17	0
nad4L	98	19	0		
nad6	176	14	0		

- Formatted: English (United Kingdom)
- Formatted: English (United Kingdom)
- Formatted: English (United Kingdom)
- Formatted: English (United Kingdom)
- Formatted: English (United Kingdom)
- Formatted: English (United Kingdom)
- Formatted: English (United Kingdom)

To further corroborate these results using a per site based approach we relied on FEL
algorithm, which uses a maximum likelihood approach to infer dN and dS substitution rates
on a per site basis for a given coding alignment and corresponding phylogeny [37]. In GL, the
datasets where sites exhibiting predominantly negative (purifying) selection outnumbered
the sites exhibiting predominantly positive (diversifying) selection were: 13PCG, *cox* family,
*atp6*, *cox1*, *cox2*, *cox3*, *nad1* and *nad4L*; and vice versa: *atp* family, *nad* family, *atp8*, *cytb*,
*nad2*, *nad3*, *nad4*, *nad5* and *nad6* (Table 2). In GP, the genes where sites exhibiting
predominantly negative (purifying) selection outnumbered the sites exhibiting
predominantly positive (diversifying) selection were: *cox* family, *cox1*, *cox3*, *nad4L* and *nad5*;
and vice versa: 13PCG, *atp* family, *nad* family, *atp6*, *atp8*, *cytb*, *cox2*, *nad1*, *nad2*, *nad3*, *nad4*
and *nad6* (Table 2). In both species, these results were observed irrespective of the
significance threshold that we applied (default = 0.1 and standard = 0.05; Additional file 5).

Formatted: Justified

Formatted: English (United Kingdom)

Formatted: English (United Kingdom)

**Table 2. Selection signals in the mitogenomes of *Gammarus lacustris* (GL) and *Gammarus pisinnus* (GP) inferred using BUSTED, aBSREL, RELAX and a**
 **per-site-based method FEL.**
 13PCG, atp, cox and nad are concatenated (non-partitioned) datasets. Sites column shows the number of codons in the alignment. p is p value. K: a
 statistically significant ($p < 0.05$) $K > 1$ indicates that selection strength has been intensified, and $K < 1$ that selection strength has been relaxed. D indicates the
 direction of selection pressure change: intensified (I) or relaxed (R). +/- is the number of sites where pervasive positive (diversifying) / negative (purifying)
 selection was identified at the 0.1 statistical threshold (default).

Data	Sites	Gammarus lacustris (GL)								Gammarus pisinnus (GP)							
		BUSTED p	aBSREL p	RELAX K LR		RELAX p D		FEL + -		BUSTED p	aBSREL p	RELAX K LR		RELAX p D		FEL + -	
13PCG	3743	0.082	0.003*	1.50	18.59	0.000*	I	24	1476	0.000*	0.001*	0.75	7.93	0.005*	R	278	0
atp	277	1.000	1.000	4.29	9.69	0.002*	I	18	0	0.056	0.085	1.00	-0.06	1.000	R	35	0
cox	1020	0.466	1.000	2.61	33.69	0.000*	I	2	501	0.520	0.500	0.81	0.01	0.941	R	5	371
nad	2055	0.161	1.000	1.00	-1.35	1.000	R	195	1	0.029*	0.102	0.78	4.24	0.039*	R	183	0
atp6	223	1.000	1.000	3.15	12.54	0.000*	I	4	157	0.506	0.420	1.00	-0.02	1.000	R	18	0
atp8	58	1.000	1.000	0.57	0.43	0.512	R	7	0	0.008*	0.002*	22.13	6.82	0.009*	I	12	0
cox1	527	0.573	1.000	3.22	16.40	0.000*	I	0	273	1.000	1.000	1.00	-0.10	1.000	R	1	177
cox2	231	0.708	1.000	1.73	3.30	0.069	I	2	81	1.000	1.000	0.86	0.18	0.670	R	8	0

Formatted: English (United Kingdom)

Formatted: English (United Kingdom)

Formatted: English (United Kingdom)

Formatted: English (United Kingdom)

Formatted: English (United Kingdom)

Formatted: English (United Kingdom)

Formatted: English (United Kingdom)

Formatted: English (United Kingdom)

Formatted: English (United Kingdom)

Formatted: English (United Kingdom)

cox3	262	0.870	0.115	3.12	10.90	0.001*	†	0	134	0.265	0.127	0.85	0.33	0.565	R	1	115
cytb	380	0.540	0.158	1.64	5.16	0.023*	†	27	0	1.000	1.000	0.32	0.12	0.733	R	35	0
nad1	313	1.000	1.000	1.27	1.53	0.216	†	2	160	0.715	0.500	0.84	0.81	0.368	R	28	0
nad2#	330	1.000	1.000	1.00	-0.47	1.000	R	37	0	1.000	0.500	0.94	0.39	0.531	R	52	0
nad3	119	1.000	1.000	2.93	3.46	0.063	†	8	0	1.000	0.179	1.00	-0.02	1.000	R	11	0
nad4	447	1.000	1.000	1.00	-0.08	1.000	R	36	0	1.000	1.000	0.82	0.79	0.374	R	37	0
nad4L	98	1.000	1.000	0.44	3.83	0.050	R	2	16	1.000	1.000	2.88	2.20	0.138	†	0	53
nad5	579	0.594	0.205	1.00	-0.35	1.000	R	64	0	0.236	0.500	1.00	-0.02	1.000	R	6	247
nad6	176	0.428	0.500	2.70	1.31	0.253	†	11	0	0.937	1.000	0.38	3.04	0.081	R	23	0

Formatted: English (United Kingdom)

Formatted: English (United Kingdom)

Formatted: English (United Kingdom)

Formatted: English (United Kingdom)

Formatted: English (United Kingdom)

Formatted: English (United Kingdom)

Formatted: English (United Kingdom)

Formatted: English (United Kingdom)

Formatted: English (United Kingdom)

Formatted: English (United Kingdom)

*Statistically significant results. #Analysis of *nad2* failed to run with the default guidance phylogenetic tree, so we ran it with the phylogeny inferred directly from the data—

40 Discussion

To test the hypothesis that ~~the speciation-mitochondrial genome~~ of *G. lacustris* may exhibit
imprints of genetic adaptation to a high-altitude environment of the Tibetan plateau, we
compared it to the mitogenome of a congeneric species inhabiting lower altitudes, ~~but not~~
that of *G. pisinnus*, as well as the dataset comprising all currently available Gammaroidea
mitogenomes. To achieve this, we ~~has been accompanied by mitogenomic adaptations to a~~
hypoxic high-altitude environment, we sequenced the complete mitochondrial genomes of
these two species, conducted comparative mitogenomic and phylogenetic analyses, and
applied a number of different tests to study the evolutionary forces that shaped these
mitogenomes.

General comparative mitogenomics

Both mitogenomes exhibited a fairly conserved architecture, standard for Gammaroidea and
Senticaudata (Figure 1, Additional file 21). A notable exception was the truncated *nad1* gene
in GL (912 bp), also shared by *Gammarus duebeni* (913 bp), whereas most other
Senticaudata (including GP) exhibited a larger size (924-933 bp). As the 3' of this gene is
poorly conserved in Senticaudata, this is an indication of its limited importance for the
functionality of this gene, so ~~we suggest that~~ it is unlikely that the truncation was adaptive,
and more likely that it is merely a consequence of a non-adaptive random mutational event
introducing a stop codon into a non-essential segment of the gene.

Strand asymmetry and skews

Strand asymmetry is a relatively common feature of organellar genomes [42-44], including

Formatted: English (United Kingdom)

Formatted: English (United Kingdom)

those of gammarids [1], but it can obstruct phylogenetic reconstruction and produce gravely
misleading phylogenetic artefacts (long-branch attraction) in cases when unrelated taxa
exhibit inverted skews [45, 46]. GP exhibited the lowest AT skew value among the available
Gammaroidea mitogenomes (-0.068) and a low GC skew value of -0.31, whereas GL
exhibited a comparatively standard AT (-0.014) and GC (-0.263) skew values. GP also
exhibited the highest A+T content (70%) among the Gammaroidea mitogenomes, whereas
GL had a comparatively low A+T content of 64.3%. This is an indication that the evolutionary
forces (adaptive or nonadaptive) that shaped the mitogenome of GP may have differed from
those of other gammarids. ~~Whereas GC skews exhibited a very intriguing pattern: most~~
~~individual genes-PCGs~~ had negative GC skews, ~~but~~ *nad1*, *nad4*, *nad4L*, *nad5* and all RNA
genes had positive GC skews. Importantly, these genes are all encoded on the minus strand,
which is in agreement with the hypothesis that ~~the likely cause for~~ strand asymmetry is
~~caused by~~ hydrolytic deamination of bases in the leading DNA leading strand when it is
single stranded, i.e. during replication and transcription [44]. The only individual outlier in
relation to this pattern was the GC skew value of tRNAs of *S. indentatus* (-0.017) (also an
outlier in the AT skew). Disregarding the minor exception of GP, gammarids exhibit relatively
consistent skews, so we can tentatively conclude that there are no indications that
mitogenomes are not a suitable tool for reconstructing the phylogeny of gammarids.

**Phylogeny and gene order**

~~In congruence with the topologies obtained using the standard ('homogeneous') models in~~
~~combination with data partitioning (Additional file 2), Pp~~araphyly of Gammaridae has ~~also~~
been reported in a relatively recent multi-gene-based revision of the Gammaroidea

Formatted: English (United Kingdom)

Formatted: English (United Kingdom)

Formatted: English (United Kingdom)

Formatted: English (United Kingdom)

Formatted: English (United Kingdom)

Formatted: English (United Kingdom)

phylogeny [47], and the split of the Baikal lake Gammaroidea into two clades has also been
~~observed before~~ reported by several studies [47-49]. However, using an almost complete
~~mitogenomic dataset (PCGRT) in combination with a 'heterogeneous' CAT-GTR model, we~~
~~managed to obtain monophyletic (polytomy notwithstanding) Gammaridae. This supports a~~
~~previous observation that CAT-GTR model appears to be more efficient at dealing with~~
~~compositional heterogeneity than data partitioning [46]. Although we tentatively accepted~~
~~this as the correct topology, this reasoning is somewhat circular, so the monophyly of this~~
~~family should be corroborated using additional datasets and data types before this issue can~~
~~be declared as resolved. The close relationship of GL and *G. duebeni* in all of our topologies~~
~~(polyphyly in CAT-GTR notwithstanding) is in disagreement with the findings of Hou & Sket~~
~~[47], and rather unexpected, as *G. duebeni* is distributed throughout the intertidal zone of~~
~~the North Atlantic region [50]. It might be merely an artefact caused by the poor sampling of~~
~~gammarid mitogenomes, resulting in unresolved polyphyletic relationships, or even~~
~~particular evolutionary pressures that have shaped the mitogenome of GL. A much denser~~
~~sampling of closely related gammarid mitogenomes would be needed to test these~~
~~hypotheses.—~~

~~While the two newly sequenced mitogenomes exhibited perfectly conserved gene~~
~~orders (among the Gammaridae), a representative of the Baikalian family Micruropodidae,~~
~~*Gmelinoides fasciatus*, exhibited —In our topology, the Baikalian family Micruropodidae~~
~~(represented by *Gmelinoides fasciatus*) was the basal taxon in the Gammaroidea clade (the~~
~~Gammaroidea dataset used in our study produced a slightly rearranged topology, with GP as~~
~~the basal taxon; Additional file 4). Intriguingly, some other studies resolved it as relatively~~

Formatted: English (United Kingdom)

Formatted: English (United Kingdom)

Formatted: English (United Kingdom)

Formatted: English (United Kingdom)

Formatted: English (United Kingdom)

Formatted: English (United Kingdom)

Formatted: English (United Kingdom)

Formatted: English (United Kingdom)

Formatted: English (United Kingdom)

Formatted: English (United Kingdom)

Formatted: Indent: First line: 1.75 pi

~~derived within the Gammaroidea clade, forming a monophyletic group with the remaining~~
~~Baikal Gammaroidea [1, 51]. Further samples and studies are needed to resolve the position~~
~~of this family. Another intriguing feature of this taxon is a highly rearranged gene order in *G.*~~
~~*fasciatus* (Figure 1), which is very unusual among the *Senticaudata* Gammaroidea. The only~~
~~species that exhibited a comparable extent of rearrangements is a Crangonyctoidea species,~~
~~*Pseudocrangonyx daejeonensis* [49].~~ The remaining two available Micruropodidae
mitogenomes are incomplete, so it is impossible to infer conclusions with confidence, but
*Linevichella vortex* appears to possess a conserved gammarid gene order, whereas
*Crypturopus tuberculatus* ~~also seems to~~ possesses a highly rearranged and unique gene
order (Figure 1)(Additional file 4). This contrast between the highly conserved architecture in
the rest of the Gammaroidea dataset and ~~(apparently)~~ highly volatile gene order in some
Micruropodidae might offer a strong ~~piece of~~ evidence in support of the hypothesis that the
evolution of mitogenomic architecture is highly discontinuous, ~~and that once a long period~~
~~of stasis in gene order and content has been punctuated by a rearrangement event, such a~~
~~destabilised mitogenome is much more likely to undergo subsequent rearrangement events,~~
~~resulting in an exponentially accelerated evolutionary rate of mitogenomic rearrangements~~
~~in certain taxonomic clades [24].~~ Therefore, it would be of greatest interest to sequence
further mitogenomes belonging to the Micruropodidae and the entire Baikal group 2 of
gammarids, as defined by Hou and Sket [47].

Formatted: English (United Kingdom)

Formatted: English (United Kingdom)

Formatted: English (United Kingdom)

Formatted: English (United Kingdom)

Formatted: English (United Kingdom)

Formatted: English (United Kingdom)

Formatted: English (United Kingdom)

Formatted: English (United Kingdom)

The close relationship of GL and *G. duebeni* clade in our results is in disagreement with
the findings of Hou & Sket [47], and rather unexpected, as *G. duebeni* is distributed
throughout the intertidal zone of the North Atlantic region [50]. We hypothesise that it

Formatted: English (United Kingdom)

Formatted: English (United Kingdom)

Formatted: English (United Kingdom)

Formatted: English (United Kingdom)

might be an artifact caused by the poor sampling of gammarid mitogenomes and/or
particular evolutionary pressures that have shaped the mitogenome of GL. A much denser
sampling of closely related gammarid mitogenomes would be needed to test these
hypotheses.—

Selective pressures

As opposed to the speciation of *G. pisinnus*, presumed to have been driven (along with a
number of other proposed endemic species) by the uplift of the Taihang mountain range
during the Miocene (23 to 5 MYA) [4, 6], the evolutionary history of Tibetan GL populations

Formatted: English (United Kingdom)

Formatted: English (United Kingdom)

is likely to have been driven largely by the Miocene uplifts in eastern Tibet and the
associated reorganization of major river drainages caused by river capture and reversal
events [36]. Given the central role of the mitochondrial genome in the energy production in

Formatted: English (United Kingdom)

Formatted: English (United Kingdom)

animals [35], we hypothesized that the mitogenome of GL may exhibit signs of adaptation
(directional selection) to life at high altitudes. Low oxygen supply presents a major challenge
for species inhabiting high-altitude habitats [31], and signs of adaptation to such an

Formatted: English (United Kingdom)

Formatted: English (United Kingdom)

environment can sometimes be observed as positive selection in genomes. For example the
Schizothoracinae fish lineage, endemic to the Tibetan Plateau, shows genome-wide
accelerated evolution relative to other fish lineages [15]. As high-altitude adaptation can

Formatted: English (United Kingdom)

Formatted: English (United Kingdom)

occur on different metabolic levels, not all of which include the mitochondrial metabolism
[31, 34], the mitochondrial genome may reflect only a very small fragment of these adaptive

Formatted: English (United Kingdom)

Formatted: English (United Kingdom)

changes. However, as the mitochondrial genome encodes 13 genes that are involved in
oxidative phosphorylation, their functioning can be affected by the environmental hypoxia
[34]. As lower atmospheric oxygen pressure is also reflected in decreased amount of

Formatted: English (United Kingdom)

Formatted: English (United Kingdom)

dissolved oxygen in water, we hypothesized that the mitogenome of GL may have
undergone episodic positive selection during the uplifting of the QTP.

In agreement with our hypothesis, we did find evidence for episodic diversifying directional
selection in GL using two different algorithms (aBSREL and BUSTED). Although, but this
result was obtained only using the entire concatenated PCG dataset. Although the BUSTED
analysis did not find evidence for positive selection in any of the GL datasets, the p value of
the 13PCG dataset was only slightly above the significance threshold (0.082). Therefore, we
can conclude that both algorithms indicate the existence of a weak positive selection signal,
which is observable only using the largest dataset. However, results of different tools are
somewhat incongruent: BUSTED and aBSREL indicate positive/relaxed evolution of the
13PCG dataset, whereas RELAX and FEL algorithms results indicate that signals of purifying
intensified selection outweigh the signals of directional selection in the mitogenome of this
species, This is expected, as purifying selection is the predominant force driving the
evolution of mitogenomes, and directional selection evolution signals are usually constrained
to specific functional sites within genes [34, 52, 53].

It should be noted here that GL possesses relatively high passive dispersal potential,
largely due to its ability to be dispersed short distances via waterfowl, which results in a
broad distribution range of this species [2, 9]. This makes it unique both among other
Tibetan life forms, which tend to be endemic [8], and other freshwater Gammarus species,
which generally have weak dispersal potential [6]. Therefore, we should not exclude a
possibility that Tibetan GL populations may be receiving a limited influx of migrants from
other locations, which may weaken the directional selection signal. As this species is

Formatted: English (United Kingdom)

Formatted: English (United Kingdom)

Formatted: Indent: First line: 1.75 pi

Formatted: English (United Kingdom)

Formatted: English (United Kingdom)

Formatted: English (United Kingdom)

Formatted: English (United Kingdom)

Formatted: English (United Kingdom)

Formatted: English (United Kingdom)

generally well-adapted to life in alpine lakes, these migrants may be able to survive in the
extreme environment of the QTP. However, molecular phylogeographic patterns indicate
(due to limited data, this is not strongly supported) that gammarid fauna of the Tibetan
plateau is probably the result of a single recent colonization event, and that genetic
admixture occurs only among different Tibetan populations [9]. Therefore, it is not a likely
scenario. Furthermore, we would still expect the migrants to exhibit adaptations to life at
high-altitudes. Future studies may attempt to sequence mitogenomes from lowland GL
populations, and ideally even those inhabiting localities exceeding elevations of 5000 m [9],
and conduct comparative analyses on the entire dataset. This can be explained by the
fact that positive selection will usually act only on a few sites for a short period of
evolutionary time, so the signal for positive selection is usually swamped by the continuous
negative selection that occurs on most sites in a gene sequence [52].

Importantly, and However, somewhat unexpectedly, all tests indicate the existence of
an even stronger signal of positive/diversifying selection, as well as overall we also
found significantly relaxed purifying selection, evidence for episodic diversifying selection
in the mitogenome of the other newly sequenced species, *G. pisinnus*. two more species
*Gammarus fossarum*, and the other newly sequenced species, GP. Intriguingly, the signal for
positive/relaxed evolution is much stronger in GP than in GL: all conducted analyses support
the existence of significant positive (or relaxed) evolutionary signal in this mitogenome,
particularly pronounced in the *atp8* gene. This fast evolution of GP is also a likely underlying
reason for the surprisingly low (for congeners) [53, 54] gene identity values between GL
and GP. The evidence for relaxation of functional evolutionary constraints (and/or positive

Formatted: English (United Kingdom)

Formatted: English (United Kingdom)

Formatted: English (United Kingdom)

Formatted: English (United Kingdom)

Formatted: English (United Kingdom)

Formatted: English (United Kingdom)

Formatted: No widow/orphan control

Formatted: English (United Kingdom), Highlight

Formatted: English (United Kingdom)

Formatted: English (United Kingdom), Highlight

Formatted: English (United Kingdom)

Formatted: English (United Kingdom)

Formatted: English (United Kingdom)

~~evolutionselection~~) was reported in the transcriptome of a subterranean *Gammarus* species
~~[55], which can. However, while it is relatively easy to be~~ explained ~~ed that observation~~ by
adaptation to a radically different (subterranean) environment. ~~However,~~ this finding is
surprising for GP, ~~and the explanation for it not as readily obvious as~~ this species does not
inhabit a unique ecological niche compared to other gammarids.

Formatted: English (United Kingdom)

Formatted: English (United Kingdom)

As mentioned above, most freshwater *Gammarus* species generally have weak dispersal
potential, so their phylogeny is strongly influenced by geological barriers, which results in
the existence of many endemic species, often with altitude-specific distribution ranges [6].

Formatted: English (United Kingdom)

Formatted: English (United Kingdom)

We hypothesise that the speciation of GP was most likely a result of fragmentation of
aquatic habitats and changes in the environment that accompanied the rapid uplifting of the
Taihang range from a vast plain that predated it [6]. This process opened new ecological

Formatted: English (United Kingdom)

Formatted: English (United Kingdom)

niches, where mutant phenotypes could be advantageous. For example, it has been
proposed that uplifting of the Lüliang and Taihang ranges promoted several speciation
events in gammarids, resulting in the existence of several (putative) gammarid species

(including GP) in a relatively narrow geographical range [6]. Such a scenario would may
explain the observed rapid evolution of the GP mitogenome, as well as the proposed
heightened speciation intensity in these mountain ranges.

Formatted: English (United Kingdom)

Formatted: English (United Kingdom)

Moreover, this may also explain the much more pronounced directional selection
pressure signal in the mitogenome of GP, compared to that of GL. The geological ages of the
two habitats are very different; the uplift of the Tibetan plateau began during the Upper

Cretaceous, some 70 MYA (million-years-ago), whereas the uplift of the Taihang mountain
range occurred during the Miocene (23 to 5 MYA) [4, 6]. Therefore, it is possible that GL

Formatted: English (United Kingdom)

Formatted: English (United Kingdom)

underwent the adaptation to the new environment much earlier than GP. Such scenario
would allow the purifying selection to swamp the signal of episodic positive evolution in GL,
but not in GP. Note that this appears incongruent with the proposal mentioned above, that
Tibetan GL populations are the result of a single recent colonization event [9], but the
authors did not propose the exact time frame for 'recent', so we cannot reject it with
certainty.

Formatted: English (United Kingdom)

Formatted: English (United Kingdom)

Furthermore, [8, 9] a possibility that the signal has been weakened by an influx of
migrants should also be considered. Because of their weak dispersal potential, the
phylogeny of freshwater Gammarus species is strongly influenced by geological barriers [6],
which results in the existence of many endemic species, often with altitude specific
distribution ranges. However, GL has a comparatively strong dispersal ability, due to its
ability to be dispersed short distances via waterfowl [2]. Therefore, it is also theoretically
possible that Tibetan GL populations may be receiving a limited influx of migrants from other
locations. As this species is generally well adapted to life in alpine lakes, these migrants may
be able to survive in the extreme environment of the QTP. Although this influx of migrants
may weaken the strength of the selection signal, they would still be expected to exhibit
adaptations to life at high altitudes. To resolve this question, it would be useful to sequence
further mitogenomes belonging to GL populations [9] from different habitats and compare
them.

Formatted: English (United Kingdom)

Formatted: English (United Kingdom), Not Highlight

Formatted: English (United Kingdom), Not Highlight

Formatted: English (United Kingdom), Not Highlight

Formatted: English (United Kingdom), Not Highlight

Formatted: English (United Kingdom), Not Highlight

[9] As accelerated mitochondrial evolution can be driven by positive directional
(adaptive) selection or by relaxation of functional constraints (nonadaptive) mutational
pressures [15, 28, 46], we should not exclude a possibility that the comparatively strong

Formatted: English (United Kingdom)

Formatted: English (United Kingdom), Not Highlight

Formatted: English (United Kingdom)

Formatted: English (United Kingdom)

Formatted: Indent: First line: 1.75 pi, No widow/orphan control

Formatted: English (United Kingdom)

Formatted: English (United Kingdom)

Field Code Changed

Formatted: English (United Kingdom)

signal in GP could be a combination of both. The exceptionally high GC skew value of the GP
mitogenome may be an indirect evidence for this, as base composition skews are believed to
be mostly driven caused by nonadaptive mutational pressures, usually driven by architectural
rearrangements or heightened metabolic rates [44-46]. Given the highly conserved
architecture of the gammarid mitogenomes, we can probably exclude architectural
rearrangements as the underlying cause [45, 46], and instead propose that it may be
associated with a (hypothetical) comparatively high metabolic rate [44, 45] in the GP.
Although it is not excessive to speculate that species living in moderate climate
environments would exhibit higher metabolic rates than species inhabiting extremely cold
environment of the Tibetan plateau, this hypothesis would first have to be tested. Specific
leaves several questions that remain open include:- Do other species described in that
same geographic range [6], exhibit heightened evolutionary rates?; and Do they exhibit
higher evolutionary rates than *Gammarus* species inhabiting other moderate environments?
Within the studied dataset, three (out of five) *Gammarus* species exhibited significant signals
of episodic diversifying/relaxed selection: GL, GP and *G. fossarum*. This is an indication that
some *Gammarus* species indeed do exhibit comparatively accelerated mitochondrial
evolution rates in comparison to other Senticaudata. As these questions warrant further
studies, it would be very interesting to sequence the mitogenomes of the remaining species
described in that geographic range [6], and test the hypothesis that these mitogenomes
should also exhibit comparatively high evolutionary rates within the entire *Gammarus*
dataset. These molecular data would also allow us to further test the extent of isolation and
gene flow between these proposed species.

Formatted: English (United Kingdom)

Formatted: English (United Kingdom)

Formatted: English (United Kingdom)

Formatted: English (United Kingdom)

Formatted: English (United Kingdom)

Formatted: English (United Kingdom)

Formatted: English (United Kingdom)

Formatted: English (United Kingdom)

Formatted: Font: Italic, English (United Kingdom)

Formatted: English (United Kingdom)

Formatted: Font: Italic, English (United Kingdom)

Formatted: English (United Kingdom)

Formatted: English (United Kingdom)

Formatted: English (United Kingdom)

Formatted: Font: Italic, English (United Kingdom)

Formatted: English (United Kingdom)

Additionally, this is a relatively newly described species [6], so data about its life history
is scarce. Several different gammarid species were described in a relatively narrow
geographical range, which led the authors to propose that uplifting of the Lüliang and
Taihang ranges (23.5 MYA) may have promoted several speciation events in gammarids [6].
~~As accelerated evolution can be driven by positive (adaptive) selection or by relaxation of~~
~~functional constraints (nonadaptive) [15, 28], the strong signal in GP could be a combination~~
~~of both.~~ We hypothesise that the most likely explanation is associated with the changes in
the environment that accompanied the rapid uplifting of the Taihang range from a vast plain
that predated it [6]. This process opened new ecological niches, where mutant phenotypes
were likely to be advantageous. Such a scenario would explain the observed features of
the mitogenome, and facilitate the proposed heightened speciation intensity [6]. It would be
very interesting to sequence the mitogenomes of the remaining six species described in that
geographic range and test the hypothesis that these mitogenomes should also exhibit
comparatively high evolutionary rates.

To get a better resolution attempt to answer the question which genes may be
especially affected by directional/relaxed selective pressures discussed above, we studied
the selection imprints on individual genes in both species, broke down the 13PCG dataset
into different subsets, which. This approach helped allowed us to identify several interesting
trends. Also, gene families and individual genes in the mitogenome of GL predominantly
exhibited varying levels of purifying selection, whereas that of GP mostly exhibited relaxed
selection. Most importantly, different data, including the gene identity values, are
indicative of high levels of purifying selection in the cox gene family (which also exhibited

Formatted: Indent: First line: 1.75 pt
Formatted: English (United Kingdom)
Formatted: English (United Kingdom)

Formatted: English (United Kingdom)
Formatted: English (United Kingdom)

Formatted: English (United Kingdom)
Formatted: English (United Kingdom)

Formatted: English (United Kingdom)
Formatted: English (United Kingdom)

Formatted: English (United Kingdom)
Formatted: English (United Kingdom)

Formatted: English (United Kingdom), Not Highlight

Formatted: English (United Kingdom)

Formatted: English (United Kingdom), Not Highlight
Formatted: English (United Kingdom)

Formatted: English (United Kingdom), Not Highlight
Formatted: English (United Kingdom)

[revised manuscript text omitted]

It has been hypothesized that the *cox* family, which catalyzes the terminal reduction of
oxygen, could be a particularly important target for high altitude adaptation [34]. Indeed,
there is evidence for adaptation to anoxia (flying at very high altitudes) in the *cox3* gene of
313 bar-headed geese [34]. However, we found strong evidence for intensive purifying selection
in the entire *cox* gene family in GL. This is not unusual, as *cox1* is commonly the most
conserved gene in animal mitogenomes in general [53, 54, 56], which includes other
gammarids (Baikal lake) [61], and other Tibetan animals (goats) [35]. It remains unclear
whether *cox* genes were unaffected by the adaptation to high altitude, or whether the signal
has been drowned out by the strong purifying selection in the meantime.

only in the *nad* gene family. Positive selection in this family was also identified in bats
[28] and in a deep-sea shrimp *Shinkaicaris leurokolos* [52]. The *nad* family encodes proteins
forming the mitochondrial proton pump, so positive evolutionary pressure in these subunits

Formatted: English (United Kingdom)
Formatted: English (United Kingdom)

Formatted: Font: Italic, English (United Kingdom)
Formatted: English (United Kingdom)

Formatted: English (United Kingdom), Not Highlight

Formatted: English (United Kingdom), Not Highlight
Formatted: English (United Kingdom), Not Highlight

Formatted: English (United Kingdom), Not Highlight
Formatted: English (United Kingdom), Not Highlight

Formatted: English (United Kingdom), Not Highlight
Field Code Changed
Formatted: English (United Kingdom), Not Highlight
Formatted: English (United Kingdom), Not Highlight
Formatted: English (United Kingdom), Not Highlight
Formatted: English (United Kingdom), Not Highlight
Formatted: English (United Kingdom), Not Highlight

Formatted: English (United Kingdom)

Formatted: English (United Kingdom), Highlight

Formatted: English (United Kingdom), Highlight
Formatted: English (United Kingdom), Highlight
Formatted: English (United Kingdom), Highlight
Formatted: English (United Kingdom), Highlight

may be associated with adjustments in the efficiency of proton pumping. This led the latter
group of authors to attribute it to adaptation to the anoxic deep sea environment [52], but
we cannot apply the same conclusion with confidence.

It has been hypothesized that the *cox* family, which catalyzes the terminal reduction of
oxygen, could be a particularly important target for high altitude adaptation [34]. Indeed,
there is evidence for adaptation to anoxia (flying at very high altitudes) in the *cox3* gene of
330 bar-headed geese [34]. However, we found strong evidence for intensive purifying selection
in the entire *cox* gene family in GL. This is not unusual, as *cox1* is commonly the most
conserved gene in animal mitogenomes in general [53, 54, 56], which includes other
gammarids (Baikal lake) [1], and other Tibetan animals (goats) [35]. It remains unclear
whether *cox* genes were unaffected by the adaptation to high altitude, or whether the signal
has been drowned out by the strong purifying selection in the meantime. A possibility that
the signal has been weakened by an influx of migrants should also be considered. Because of
their weak dispersal potential, the phylogeny of freshwater *Gammarus* species is strongly
influenced by geological barriers [6], which results in the existence of many endemic species,
often with altitude specific distribution ranges. However, GL has a comparatively strong
dispersal ability, due to its ability to be dispersed short distances via waterfowl [2].
Therefore, it is also theoretically possible that Tibetan GL populations may be receiving a
limited influx of migrants from other locations. As this species is generally well adapted to
life in alpine lakes, these migrants may be able to survive in the extreme environment of the
QTP. Although this influx of migrants may weaken the strength of the selection signal, they
would still be expected to exhibit adaptations to life at high altitudes. To resolve this

Formatted: English (United Kingdom), Highlight

Formatted: English (United Kingdom), Highlight

Formatted: English (United Kingdom)

Formatted: English (United Kingdom), Highlight

Formatted: English (United Kingdom), Highlight

Formatted: English (United Kingdom), Highlight

Formatted: English (United Kingdom), Highlight

Formatted: English (United Kingdom), Highlight

Formatted: English (United Kingdom), Highlight

Field Code Changed

Formatted: English (United Kingdom), Highlight

Formatted: English (United Kingdom), Highlight

Field Code Changed

Formatted: English (United Kingdom), Highlight

Formatted: English (United Kingdom), Highlight

Formatted: English (United Kingdom), Highlight

Formatted: English (United Kingdom), Highlight

Formatted: English (United Kingdom), Highlight

Formatted: English (United Kingdom), Highlight

Formatted: English (United Kingdom), Highlight

question, it would be useful to sequence further mitogenomes belonging to GL populations
from different habitats and compare them.

Formatted: English (United Kingdom)

The two genes in the *atp* family exhibited different pressures in GL: purifying in *atp6*
and relaxed in *atp8*. In GP, however, we found strong evidence for intensified selection in
the *atp8* (only the FEL analysis did not find evidence for significantly intensified selection).
Relaxed selection in *atp8* is not surprising as it is a very small gene that generally appears to
be under lesser evolutionary constraints, so it is even missing from the mitogenomes of
many metazoan taxa [1, 21, 53, 54, 56, 60]. Although this is not a unique phenomenon, for
example significant adaptive variation was identified in this gene in cephalopods [62], it is
very intriguing and deserves further studies.

Formatted: English (United Kingdom), Highlight

Formatted: English (United Kingdom), Highlight

Formatted: English (United Kingdom), Highlight

Formatted: English (United Kingdom), Highlight

Formatted: English (United Kingdom), Highlight

Formatted: English (United Kingdom)

Conclusions

We identified the existence of directional evolution signal in the mitogenome of GL, which
indirectly supports our working hypothesis, but a surprising finding is that the signal of
relaxed/directional selective pressures is much stronger in the mitogenome of GP. This is
surprising, as GP does not inhabit a unique ecological niche compared to other gammarids.
We hypothesise that the speciation of GP was most likely a result of fragmentation of
aquatic habitats and changes in the environment that accompanied the rapid uplifting of the
Taihang range from a vast plain that predated it, which opened new ecological niches, where
mutant phenotypes could be advantageous. Such scenario may explain the observed rapid
evolution of the GP mitogenome, as well as the proposed heightened speciation intensity in
these mountain ranges. We hypothesise that much older age of the Tibetan plateau may

have allowed the purifying selection to swamp the signal of episodic positive evolution in GL,
whereas the relative recency of its speciation may explain the stronger signal in GP. We
propose that this hypothesis can be tested by sequencing the mitogenomes of the remaining
six (proposed) species populating the same narrow geographic range. As all of these species
are believed to be a result of these same geological events, their mitogenomes should
exhibit similar evolutionary footprints. Furthermore, as we found indications that some
Gammarus species exhibit comparatively accelerated mitochondrial evolution rates in
comparison to other Senticaudata, we propose that future studies should not exclude the
existence of unusually strong nonadaptive mutational pressures in their mitogenomes.

We Although the architecture of the two newly sequenced mitogenomes is very similar,
gene identity values were surprisingly low for congeners [53, 54]. This is likely to be a
consequence not only of the adaptation of GL to environmental changes associated with the
life at high altitudes, but also of the apparent mix of positive and relaxed evolutionary
pressures on the mitogenome of GP. We can conclude that our results provide only a weak
found support for our working hypothesis (directional evolution signal in the mitogenome of
GL).

However, a. A much more surprising finding is that the signal of relaxed/directional
selective pressures is much stronger in evidence for a combination of positive and relaxed
evolutionary pressures shaping the mitogenome of GP. We hypothesise that the most likely
explanation is associated with the changes in the environment that accompanied the rapid
uplifting of the Taihang mountains, which. appears to have promoted several speciation
events in gammarids [6]. As the speciation of *G. pisinnus* is presumed to have occurred

Formatted: Indent: First line: 0 pi

Formatted: English (United Kingdom)

Formatted: English (United Kingdom)

Formatted: English (United Kingdom)

Formatted: English (United Kingdom)

relatively recently [6], the signal of episodic positive evolution associated with
environmental changes that eventually resulted in its speciation may have remained much
clearer than in GL. We propose that this hypothesis can be tested by sequencing the
mitogenomes of the remaining six (proposed) species populating the same narrow
geographic range. As all of these species are believed to be a result of these same geological
events, their mitogenomes should exhibit similar evolutionary footprints. This is an
indication that some *Gammarus* species indeed do exhibit comparatively accelerated
mitochondrial evolution rates in comparison to other Senticaudata. Finally, we propose that
different ages of the two species are the most likely underlying reason for the stronger
positive selection signal in GP. As purifying (negative) selection is the predominant force
driving the evolution of mitogenomes [34, 52], a much longer time that has passed since the
uplift of the QTP (70 MYA) would allow the purifying selection to swamp the signal of
episodic positive evolution in the mitogenome of *G. lacustris*. As the speciation of *G. pisinnus*
is presumed to have occurred much later, 6.4 MYA [6], the signal of episodic positive
evolution associated with environmental changes that eventually resulted in its speciation
has remained much clearer than in GL.

[9]

Methods

Sampling

Formatted: English (United Kingdom)

Formatted: English (United Kingdom)

Formatted: English (United Kingdom)

Formatted: English (United Kingdom)

Formatted: English (United Kingdom)

Formatted: English (United Kingdom)

Formatted: English (United Kingdom)

Formatted: Widow/Orphan control

Formatted: English (United Kingdom)

*Gammarus lacustris* samples were collected at the altitude of ≈ 3400 m, in the Cuomujiri
lake (E $26^{\circ} 52'$ - $30^{\circ} 40'$, N $92^{\circ} 09'$ - $98^{\circ} 47'$) near the Nyingchi city in Tibet (Figure 2).
*Gammarus pisinnus* samples were collected at the altitude of ≈ 510 m, from the Yellow
River near the Xian City, Shanxi province, China (E $108^{\circ} 54'$, N $34^{\circ} 16'$). Both samplings
were conducted in August 2017 using fine-meshed hand nets. Approximately 50Fifty
specimens were collected at each site, immediately preserved in 95% ethanol in the field,
and long-term stored at -20°C for later use. Samples were morphologically identified under
the light microscope as described before [4, 6]. As gammarids are non-protected
invertebrates, no permits were needed for sampling and sample processing.

Formatted: English (United Kingdom)

Formatted: English (United Kingdom)

Formatted: English (United Kingdom)

Formatted: English (United Kingdom)

**Figure 2. Approximate sampling localities for *Gammarus pisinnus* (GP) and *Gammarus***
 ***lacustris* (GL).**

**DNA extraction, amplification and sequencing**

These steps were conducted as described before [24, 63, 64], so only an overview is
 provided. DNA was isolated from a complete single specimen (Aidlab DNA extraction
 kit, Aidlab Biotechnologies, Beijing, China) of each species after rinsing them in
 distilled water. Conserved fragments of several genes were amplified and sequenced using
 degenerate primer pairs (Additional file 623) designed using available gammarid
 orthologues/mitogenomes as templates. These gene fragments were then used to design
 specific primers (overlapping by approximately 100 bp) for amplification and sequencing of
 the whole mitogenome. PCR reaction mixture (20 µl): 7.4 µl ddH₂O, 10 µl 2×PCR buffer

Formatted: English (United Kingdom)

Formatted: English (United Kingdom)

Formatted: English (United Kingdom)

Formatted: English (United Kingdom)

(Takara, Dalian, China), 0.6 µl of each primer, 0.4 µl rTaq polymerase (250 U, Takara), and 1
µl DNA template. Amplification: 98°C/2min, 40 cycles of 98°C/10s, 50°C/15s, 68°C 1min/kb,
and finally 68°C/10min. PCR products were sequenced bi-directionally using the Sanger
method and the same set of primers.

Sequence assembly and analysis

~~These steps, Mitogenomes were assembled, annotated and comparative analyses, were~~
~~also conducted following the methodology, largely as~~ described before in more detail

[24, 63, 64], [24, 64]. Briefly, sequenced fragments were quality-proofed by visually

inspecting the electropherograms and queried against the GenBank using BLAST to confirm

that the amplicon is the target sequence. During the manual assembly, conducted with the

help of DNASTar v7.1 [65], we made sure that the overlaps were identical, mitogenome

circular, and that no numts [66] were incorporated. ORFs for protein-coding genes (PCGs)

were located using DNASTar, and manually fine-tuned via a comparison with available

gammarid orthologues using BLAST and BLASTx. tRNAscan [67] and ARWEN [68] were used

to identify tRNAs. PhyloSuite [69] was used to parse and extract mitogenomes annotated in

Microsoft Word documents, as well as to create the files for submission to GenBank and

generate all comparative mitogenomic architecture tables. Following the arguments put

forward ~~in~~ before [70], instead of referring to strands as heavy and light, we used plus (+) and

(-) to denominate the strands encoding a majority and minority of genes respectively. The

mitogenomes are available from the GenBank repository under the accession numbers

MK354235 (*Gammarus lacustris*) and MK354236 (*Gammarus pisinnus*).

Formatted: English (United Kingdom)

Formatted: English (United Kingdom)

Formatted: English (United Kingdom)

Formatted: English (United Kingdom)

Formatted: English (United Kingdom)

Formatted: English (United Kingdom)

Formatted: English (United Kingdom)

Formatted: English (United Kingdom)

Formatted: English (United Kingdom)

Formatted: English (United Kingdom)

Formatted: English (United Kingdom)

Formatted: English (United Kingdom)

Formatted: English (United Kingdom)

Formatted: English (United Kingdom)

Formatted: English (United Kingdom)

Formatted: English (United Kingdom)

**Comparative mitogenomic and phylogenetic analyses**

To minimize the impact of compositional heterogeneity, for phylogenetic and selection
pressure analyses we used only all 13 mitogenomes belonging to the superfamily
Gammaroidea available in the GenBank, plus *Metacrangonyx remyi* (Senticaudata:
*Hadzioidea*) [38] as the outgroup. As some Gammaroidea mitogenomes in that dataset were

Formatted: English (United Kingdom)
Formatted: English (United Kingdom)

incomplete, for comparative analyses we retrieved all 15 available mitogenomes belonging
to the infraorder Senticaudata (class Malacostraca: order Amphipoda) available in the
RefSeq database (Additional file 1; alignments available in the Additional file 4). We

Formatted: English (United Kingdom)
Formatted: English (United Kingdom)

retrieved all 15 available mitogenomes belonging to the infraorder Senticaudata (class
Malacostraca: order Amphipoda) available in the RefSeq database. To stabilize the topology,
we used three other crustacean species as outgroups for the phylogenetic analysis: a basal
crustacean *Limulus polyphemus* (class Merostomata) [71], *Squilla mantis* (Malacostraca:
order Stomatopoda) [72], and *Cymothoa indica* (Malacostraca: order Isopoda) [18].

Formatted: English (United Kingdom)
Formatted: English (United Kingdom)
Formatted: English (United Kingdom)
Formatted: English (United Kingdom)
Formatted: English (United Kingdom)

PhyloSuite was used to batch-download all selected mitogenomes from the GenBank,
extract genomic features, translate genes into amino acid sequences, and
semi-automatically re-annotate ambiguously annotated tRNA genes with the help of the
ARWEN output. The GenBank taxonomy was automatically replaced with the WoRMS
database taxonomy using PhyloSuite, as the latter tends to be more up to date [73].

Formatted: English (United Kingdom)
Formatted: English (United Kingdom)

To test the stability of topology, we ~~We~~ conducted phylogenetic analyses and using
different methods and datasets. Datasets used were: amino acid sequences (AAs) of all 13
concatenated PCGs (PCG concatenated)AAs), nucleotides of concatenated 13 PCGs
(PCG_NUC), and nucleotides of all mitochondrial genes (PCGRT: PCGs+rRNAs+tRNAs). Genes

~~were aligned in batches with MAFFT [74], using '--auto' strategy, and 'codon' alignment~~
~~mode for nucleotides and 'normal' mode for amino acid sequences of PCGs; whereas all RNA~~
~~genes were aligned using Q-INS-i algorithm, which takes secondary structure information~~
~~into account. PartitionFinder [75] was used to infer the best partitioning strategy and select~~
~~best-fit evolutionary models for each partition using the greedy algorithm and AICC criterion~~
~~(Additional file 4). To infer the phylogeny, we used two standard methods, both of which~~
~~assume compositional homogeneity of data (herein referred to as 'homogeneous' models),~~
~~Maximum likelihood (ML) conducted using IQ-TREE [76] and Bayesian Inference (BI)~~
~~conducted using MrBayes 3.2.6 [77]. Maximum likelihood (ML) analysis phylogeny was~~
~~conducted inferred using IQ-TREE [76] with 100 standard bootstraps, and Bayesian~~
~~inference (BI) analysis was conducted performed in MrBayes 3.2.6 [77] with default~~
~~settings, 2×10⁶ metropolis-coupled MCMC generations (analysis was stopped as the average~~
~~standard deviation of split frequencies =was sufficiently low: 0.0024), and the dataset~~
~~divided into three partitions. All these analyses were conducted with the help of PhyloSuite.~~
~~As compositional heterogeneity can compromise phylogenetic analyses, phylogeny was also~~
~~inferred using the CAT-GTR site mixture model implemented in PhyloBayes-MPI 1.7a, which~~
~~allows for site-specific rates of mutation (herein referred to as 'heterogeneous' model)~~
~~[78]. This analysis was conducted on using several plug-in programs integrated into~~
~~PhyloSuite (Flowchart mode). The 13 sequences were aligned in batches with MAFFT [74]~~
~~using '--auto' strategy and normal alignment mode for amino acids. Ambiguously aligned~~
~~fragments of these alignments were removed in batches using Gblocks [79] with the~~
~~following parameter settings: minimum number of sequences for a conserved/flank position~~

Formatted: English (United Kingdom)

Formatted: English (United Kingdom)

Formatted: English (United Kingdom)

Formatted: English (United Kingdom)

Formatted: English (United Kingdom)

Formatted: English (United Kingdom)

Formatted: English (United Kingdom)

Formatted: English (United Kingdom)

Formatted: English (United Kingdom)

Field Code Changed

Formatted: English (United Kingdom)

Formatted: English (United Kingdom)

Formatted: English (United Kingdom)

Formatted: English (United Kingdom)

Formatted: English (United Kingdom)

Formatted: English (United Kingdom)

Formatted: English (United Kingdom)

Formatted: English (United Kingdom)

Formatted: English (United Kingdom)

~~11/11, maximum number of contiguous non conserved positions 8, minimum length of a~~
~~block 10, allowed gap positions 'with half'. ModelFinder [75, 80] was used to select the~~
~~best fit evolutionary models for each gene (partition model) [81] using the BIC criterion.~~
~~Maximum likelihood (ML) phylogeny was inferred using IQ-TREE [76] with 100 standard~~
~~bootstraps. Bayesian Inference (BI) analysis was performed in MrBayes 3.2.6 [77] with~~
~~default settings, 2x10⁶ metropolis-coupled MCMC generations (average standard deviation~~
~~of split frequencies = 0.0024), and the dataset divided into three partitions. [78] beta version~~
~~of the Cipres server (<https://cushion3.sdsc.edu/portal2/tools.action>) [82], with default~~
~~parameters (burnin = 500, invariable sites automatically removed from the alignment, two~~
~~MCMC chains), -dgam set to 8, and the analysis was automatically stopped when the~~
~~conditions considered to indicate a good run were reached: maxdiff < 0.1 and minimum~~
~~effective size > 300 (PhyloBayes manual). Phylograms and gene orders were visualized in~~
~~iTOL [83] using files generated by PhyloSuite.~~

Formatted: English (United Kingdom)

Formatted: English (United Kingdom)

Formatted: English (United Kingdom)

Formatted: English (United Kingdom)

Formatted: English (United Kingdom)

Field Code Changed

Formatted: English (United Kingdom)

Formatted: English (United Kingdom)

Formatted: English (United Kingdom)

Formatted: English (United Kingdom)

Formatted: English (United Kingdom)

Formatted: English (United Kingdom)

Formatted: English (United Kingdom)

Formatted: English (United Kingdom)

Formatted: English (United Kingdom)

Formatted: English (United Kingdom)

Formatted: English (United Kingdom)

Selection pressure analyses

~~We used nucleotides of 13 PCGs (aligned using the codon mode; Additional file 4) and~~
~~PhyloBayes topology (Figure 1) as the .To minimize the impact of mutational saturation, for~~
~~these analyses we removed distantly removed species, and used all the 13 mitogenomes~~
~~belonging to the superfamily Gammaroidea available in the GenBank, plus *Metaerangonyx*~~
~~*remyi* (Hadzioidea) [38] as the outgroup. A guidance tree (Additional file 4) for the~~
~~downstream analyses was produced with MrBayes as described above, but we used~~
~~nucleotides of 13 PCGs instead of AAs and aligned them using the codon mode. The~~
~~numbers of segregating sites numbers of synonymous and nonsynonymous mutations were~~

Formatted: English (United Kingdom)

Formatted: English (United Kingdom)

estimated using ~~the~~ DnaSP [84]. We tested a number of different evolutionary hypotheses
using tools available from the DataMonkey webserver [85]; aBSREL to test if positive
selection has occurred on a proportion of branches [39], BUSTED to test for a gene-wide (not
site-specific) positive selection [40], RELAX to detect relaxed selection [41], and FEL to infer
~~nonsynonymous~~nonsynonymous (dN) and synonymous (dS) substitution rates on a per-site
basis [37].

Formatted: English (United Kingdom)
Formatted: English (United Kingdom)
Formatted: English (United Kingdom)
Formatted: English (United Kingdom)
Formatted: English (United Kingdom)
Formatted: English (United Kingdom)
Formatted: English (United Kingdom)
Formatted: English (United Kingdom)
Formatted: English (United Kingdom)
Formatted: English (United Kingdom)

Data Availability

*The two new mitogenomes are available from the GenBank repository under accession*
*numbers MK354235 (Gammarus lacustris) and MK354236 (Gammarus pisinnus), and all*
*other data generated or analysed during this study are included in this published article and*
*its supplementary information files.*

Competing Interests

*We have no competing interests.*

Authors' Contributions

*SS and XG participated in the conceptualization and design of the study, collection of samples,*
*lab work, data analysis, and drafting the manuscript; YW participated in the collection of*
*samples, lab work, data analysis, and critically revised the manuscript for important*
*intellectual content; IJ participated in the design of the study, data analysis, interpretation*

*and visualisation, and drafting the manuscript; JZ, SM, KAAG and FAM participated in the*
*molecular lab work, data analysis and visualization, and critically revised the manuscript for*
*important intellectual content; HF conceived and coordinated the study, and critically revised*
*the manuscript for important intellectual content. All authors approved the final version and*
*agree to be accountable for all aspects of the work.*

Formatted: English (United Kingdom)

**Funding**

*This work was supported by National Natural Science Foundation of China [No. 31402280];*
*the New Varieties Creation Major Project in Jiangsu Province [PZCZ201745]; the Science &*
*Technology Supporting Program of Jiangsu Province [BE2016308]; the China Agriculture*
*Research System-48 [CARS-48]; and the Deanship of Scientific Research at King Saud*
*University [1440-0138RGP-1435-012].*

**Research Ethics**

Does not apply.—

**Animal Ethics**

*This study was approved by the Animal Care and Use Committee of the Centre for Applied*
*Aquatic Genomics at Chinese Academy of Fishery Sciences.*

**Permission to carry out fieldwork**

Formatted: English (United Kingdom)

*As sampling was conducted on unprotected invertebrates and on public land, fieldwork*
*permits were not required.*

**Acknowledgements**

*The authors would like to express their sincere appreciation to the Deanship of Scientific*
*Research at King Saud University for its funding of this research through the Research Group*
*Project ~~No. 1440-0138~~No. RGP-1435-012.*

**References**

- Romanova, E. V., Aleoshin, V. V., Kamal'tynov, R. M., Mikhailov, K. V., Logacheva, M. D.,
Sirotinina, E. A., Gornov, A. Y., Anikin, A. S., Sherbakov, D. Y. Evolution of mitochondrial genomes
in Baikalian amphipods. *BMC genomics* 2016:1016.
- Vainola, R., Witt, J. D. S., Grabowski, M., Bradbury, J. H., Jazdzewski, K., Sket, B. 2008 Global
diversity of amphipods (Amphipoda; Crustacea) in freshwater. *Hydrobiologia*. **595**, 241-255.
- Costa, F. O., Henzler, C. M., Lunt, D. H., Whiteley, N. M., Rock, J., Biodiversity. 2009 Probing
marine Gammarus (Amphipoda) taxonomy with DNA barcodes. *Systematics and Biodiversity*. **7**,
365-379.
- Hou, Z., Li, S. 2018 Four new Gammarus species from Tibetan Plateau with a key to Tibetan
freshwater gammarids (Crustacea, Amphipoda, Gammaridae). *Zookeys*. **747**, 1-40.
(10.3897/zookeys.747.21999)
- Qi, D., Chao, Y., Guo, S., Zhao, L., Li, T., Wei, F., Zhao, X. 2012 Convergent, Parallel and
Correlated Evolution of Trophic Morphologies in the Subfamily Schizothoracinae from the
Qinghai-Tibetan Plateau. *PLOS ONE*. **7**, e34070. (10.1371/journal.pone.0034070)
- Hou, Z., Li, J., Li, S. 2014 Diversification of low dispersal crustaceans through mountain uplift: a
case study of Gammarus (Amphipoda: Gammaridae) with descriptions of four novel species.
*Zoological Journal of the Linnean Society*. **170**, 591–633. (doi:10.1111/zoj.12119)
- Wu, Z., Barosh, P. J., Wu, Z., Hu, D., Zhao, X., Ye, P. 2008 Vast early Miocene lakes of the central
Tibetan Plateau. *Geological Society of America Bulletin*. **120**, 1326-1337.
- Clewing, C., Albrecht, C., Wilke, T. 2016 A Complex System of Glacial Sub-Refugia Drives
Endemic Freshwater Biodiversity on the Tibetan Plateau. *PLOS ONE*. **11**, e0160286.
- Clewing, C., Wilke, T., Ilge, A., Albrecht, C. 2016 Phylogenetic patterns of freshwater amphipods
inhabiting the Tibetan Plateau. *Crustaceana*. **89**, 239–249.
(<https://doi.org/10.1163/15685403-00003518>)
- Qiu, Q., Zhang, G., Ma, T., Qian, W., Wang, J., Ye, Z., Cao, C., Hu, Q., Kim, J., Larkin, D. M.
2012 The yak genome and adaptation to life at high altitude. *Nature Genetics*. **44**, 946-949.
- 11 Qu, Y., Zhao, H., Han, N., Zhou, G., Song, G., Gao, B., Tian, S., Zhang, J., Zhang, R., Meng, X.
2013 Ground tit genome reveals avian adaptation to living at high altitudes in the Tibetan plateau.
*Nature Communications*. **4**, 2071.
- 12 Gou, X., Wang, Z., Li, N., Qiu, F., Xu, Z., Yan, D., Yang, S., Jia, J., Kong, X., Wei, Z., *et al.* 2014

Formatted: English (United Kingdom)

Whole-genome sequencing of six dog breeds from continuous altitudes reveals adaptation to
high-altitude hypoxia. *Genome Research*. **24**, 1308-1315. (10.1101/gr.171876.113)

13 Bigham, A., Bauchet, M., Pinto, D., Mao, X., Akey, J. M., Mei, R., Scherer, S. W., Julian, C. G.,
Wilson, M. J., Herráez, D. L., *et al.* 2010 Identifying Signatures of Natural Selection in Tibetan and
Andean Populations Using Dense Genome Scan Data. *PLOS Genetics*. **6**, e1001116.
(10.1371/journal.pgen.1001116)

14 Zhang, D., Yu, M., Hu, P., Peng, S., Liu, Y., Li, W., Wang, C., He, S., Zhai, W., Xu, Q., *et al.* 2017
Genetic Adaptation of Schizothoracine Fish to the Phased Uplifting of the Qinghai–Tibetan Plateau.
*G3: Genes, Genomes, Genetics*. **7**, 1267-1276. (10.1534/g3.116.038406)

15 Guan, L., Chi, W., Xiao, W., Chen, L., He, S. 2014 Analysis of hypoxia-inducible factor alpha
polyploidization reveals adaptation to Tibetan plateau in the evolution of schizothoracine fish. *BMC*
*Evolutionary Biology*. **14**, 192. (10.1186/s12862-014-0192-1)

16 Scheinfeldt, L. B., Tishkoff, S. A. 2010 Living the high life: high-altitude adaptation. *Genome Biol.*
**11**, 133. (10.1186/gb-2010-11-9-133)

17 Bourguignon, T., Tang, Q., Ho, S. Y. W., Juna, F., Wang, Z., Arab, D. A., Cameron, S. L., Walker,
614 J., Rentz, D., Evans, T. A., *et al.* 2018 Transoceanic Dispersal and Plate Tectonics Shaped Global
Cockroach Distributions: Evidence from Mitochondrial Phylogenomics. *Molecular Biology and*
*Evolution*. **35**, 970–983. (10.1093/molbev/msy013)

18 Zou, H., Jakovlic, I., Zhang, D., Chen, R., Mahboob, S., Al-Ghanim, K. A., Al-Misned, F., Li, W.,
X., Wang, G. T. 2018 The complete mitochondrial genome of *Cymothoa indica* has a highly rearranged
gene order and clusters at the very base of the Isopoda clade. *PLoS One*. **13**, e0203089.
(10.1371/journal.pone.0203089)

19 Perkins, E. M., Donnellan, S. C., Bertozzi, T., Whittington, I. D. 2010 Closing the mitochondrial
circle on paraphyly of the Monogenea (Platyhelminthes) infers evolution in the diet of parasitic
flatworms. *Int J Parasitol*. **40**, 1237-1245. (10.1016/j.ijpara.2010.02.017)

20 Li, H., Leavengood, J. M., Jr., Chapman, E. G., Burkhardt, D., Song, F., Jiang, P., Liu, J., Zhou, X.,
Cai, W. 2017 Mitochondrial phylogenomics of Hemiptera reveals adaptive innovations driving the
diversification of true bugs. *Proc Biol Sci*. **284**. (10.1098/rspb.2017.1223)

21 Liu, Z. Q., Liu, Y. F., Kuermanali, N., Wang, D. F., Chen, S. J., Guo, H. L., Zhao, L., Wang, J. W.,
Han, T., Wang, Y. Z., *et al.* 2018 Sequencing of complete mitochondrial genomes confirms
synonymization of *Hyalomma asiaticum asiaticum* and *kozlovi*, and advances phylogenetic hypotheses
for the Ixodidae. *PLoS One*. **13**, e0197524. (10.1371/journal.pone.0197524)

22 Huyse, T., Buchmann, K., Littlewood, D. T. 2008 The mitochondrial genome of *Gyrodactylus*
*derjavinoideus* (Platyhelminthes: Monogenea)--a mitogenomic approach for *Gyrodactylus* species and
strain identification. *Gene*. **417**, 27-34.

23 Lan, T., Gill, S., Bellemain, E., Bischof, R., Nawaz, M. A., Lindqvist, C. 2017 Evolutionary history
of enigmatic bears in the Tibetan Plateau-Himalaya region and the identity of the yeti. *Proc Biol Sci*.
**284**, 20171804. (10.1098/rspb.2017.1804)

24 Zou, H., Jakovlić, I., Chen, R., Zhang, D., Zhang, J., Li, W. X., Wang, G. T. 2017 The complete
mitochondrial genome of parasitic nematode *Camallanus cotti*: extreme discontinuity in the rate of
mitogenomic architecture evolution within the Chromadorea class. *BMC Genomics*. **18**, 840.
(10.1186/s12864-017-4237-x)

Ballard, J. W. O., Pichaud, N., Fox, C. 2014 Mitochondrial DNA: more than an evolutionary
bystander. *Functional Ecology*. **28**, 218-231. (10.1111/1365-2435.12177)

Wolff, J. N., Ladoukakis, E. D., Enríquez, J. A., Dowling, D. K. 2014 Mitonuclear interactions:
evolutionary consequences over multiple biological scales. *Philosophical Transactions of the Royal*
*Society B: Biological Sciences*. **369**,

Boore, J. L., Fuerstenberg, S. I. 2008 Beyond linear sequence comparisons: the use of genome-level
characters for phylogenetic reconstruction. *Philosophical Transactions of the Royal Society B:*
*Biological Sciences*. **363**, 1445. (10.1098/rstb.2007.2234)

Botero-Castro, F., Tilak, M. K., Justy, F., Catzeflis, F., Delsuc, F., Douzery, E. J. P. 2018 In Cold
Blood: Compositional Bias and Positive Selection Drive the High Evolutionary Rate of Vampire Bats
Mitochondrial Genomes. *Genome Biol Evol.* **10**, 2218-2239. (10.1093/gbe/evy120)

Peng, Y., Yang, Z., Zhang, H., Cui, C., Qi, X., Luo, X., Tao, X., Wu, T., Ouzhuluobu, Basang, *et al.*
2011 Genetic Variations in Tibetan Populations and High-Altitude Adaptation at the Himalayas.
*Molecular Biology and Evolution*. **28**, 1075-1081. (10.1093/molbev/msq290)

Li, Y., Wu, D.-D., Boyko, A. R., Wang, G.-D., Wu, S.-F., Irwin, D. M., Zhang, Y.-P. 2014
Population Variation Revealed High-Altitude Adaptation of Tibetan Mastiffs. *Molecular Biology and*
*Evolution*. **31**, 1200-1205. (10.1093/molbev/msu070)

Beall, C. M. 2007 Two routes to functional adaptation: Tibetan and Andean high-altitude natives.
*Proceedings of the National Academy of Sciences*. **104**, 8655-8660. (10.1073/pnas.0701985104)

Yang, L., Wang, Y., Zhang, Z., He, S. 2015 Comprehensive Transcriptome Analysis Reveals
Accelerated Genic Evolution in a Tibet Fish, *Gymnodiptychus pachycheilus*. *Genome Biology and*
*Evolution*. **7**, 251-261. (10.1093/gbe/evu279)

Sluis, V. D., O, E., Bauerschmitt, H., Becker, T., Mielke, T., Frauenfeld, J., Berninghausen, O.,
Neupert, W., Herrmann, J. M., Beckmann, R. 2015 Parallel Structural Evolution of Mitochondrial
Ribosomes and OXPHOS Complexes. *Genome Biology and Evolution*. **7**, 1235-1251.
(10.1093/gbe/evv061)

Scott, G. R., Schulte, P. M., Egginton, S., Scott, A. L. M., Richards, J. G., Milsom, W. K. 2011
Molecular Evolution of Cytochrome c Oxidase Underlies High-Altitude Adaptation in the Bar-Headed
Goose. *Molecular Biology and Evolution*. **28**, 351-363. (10.1093/molbev/msq205)

Hassanin, A., Ropiquet, A., Couloux, A., Cruaud, C. 2009 Evolution of the mitochondrial genome
in mammals living at high altitude: New insights from a study of the tribe Caprini (Bovidae,
Antilopinae). *Journal of Molecular Evolution*. **68**, 293-310. (10.1007/s00239-009-9208-7)

36 Ma, X., Kang, J., Chen, W., Zhou, C., He, S. 2015 Biogeographic history and high-elevation
adaptations inferred from the mitochondrial genome of Glyptosternoid fishes (Sisoridae, Siluriformes)
from the southeastern Tibetan Plateau. *BMC Evolutionary Biology*. **15**, 233.
(10.1186/s12862-015-0516-9)

Kosakovsky Pond, S. L., Frost, S. D. W. 2005 Not So Different After All: A Comparison of
Methods for Detecting Amino Acid Sites Under Selection. *Molecular Biology and Evolution*. **22**,
1208-1222. (10.1093/molbev/msi105)

Bauzá-Ribot, Maria M., Juan, C., Nardi, F., Oromí, P., Pons, J., Jaume, D. 2012 Mitogenomic
Phylogenetic Analysis Supports Continental-Scale Vicariance in Subterranean Thalassoid Crustaceans.
*Current Biology*. **22**, 2069-2074. (<https://doi.org/10.1016/j.cub.2012.09.012>)

Smith, M. D., Wertheim, J. O., Weaver, S., Murrell, B., Scheffler, K., Kosakovsky Pond, S. L.
2015 Less Is More: An Adaptive Branch-Site Random Effects Model for Efficient Detection of
Episodic Diversifying Selection. *Molecular Biology and Evolution*. **32**, 1342-1353.
(10.1093/molbev/msv022)

Murrell, B., Weaver, S., Smith, M. D., Wertheim, J. O., Murrell, S., Aylward, A., Eren, K., Pollner,
688 T., Martin, D. P., Smith, D. M., *et al.* 2015 Gene-wide identification of episodic selection. *Molecular*
*biology and evolution*. **32**, 1365-1371. (10.1093/molbev/msv035)

Wertheim, J. O., Murrell, B., Smith, M. D., Kosakovsky Pond, S. L., Scheffler, K. 2015 RELAX:
Detecting Relaxed Selection in a Phylogenetic Framework. *Molecular Biology and Evolution*. **32**,
820-832. (10.1093/molbev/msu400)

Wei, S.-J., Shi, M., Chen, X.-X., Sharkey, M. J., Achterberg, C. v., Ye, G.-Y., He, J.-H. 2010 New
Views on Strand Asymmetry in Insect Mitochondrial Genomes. *PLOS ONE*. **5**, e12708.
(10.1371/journal.pone.0012708)

Hassanin, A. 2006 Phylogeny of Arthropoda inferred from mitochondrial sequences: Strategies for
limiting the misleading effects of multiple changes in pattern and rates of substitution. *Molecular*
*Phylogenetics and Evolution*. **38**, 100–116. (10.1016/j.ympev.2005.09.012)

Reyes, A., Gissi, C., Pesole, G., Saccone, C. 1998 Asymmetrical directional mutation pressure in
the mitochondrial genome of mammals. *Molecular Biology and Evolution*. **15**, 957–966.
(10.1093/oxfordjournals.molbev.a026011)

Hassanin, A., Léger, N., Deutsch, J. 2005 Evidence for multiple reversals of asymmetric mutational
constraints during the evolution of the mitochondrial genome of metazoa, and consequences for
phylogenetic inferences. *Systematic Biology*. **54**, 277–298. (10.1080/10635150590947843)

Zhang, D., Zou, H., Hua, C.-J., Li, W.-X., Mahboob, S., Al-Ghanim, K. A., Al-Misned, F., Jakovlić,
I., Wang, G.-T. 2019 Mitochondrial Architecture Rearrangements Produce Asymmetrical Nonadaptive
Mutational Pressures That Subvert the Phylogenetic Reconstruction in Isopoda. *Genome Biology and*
*Evolution*. **11**, 1797-1812. (10.1093/gbe/evz121)

Hou, Z., Sket, B. 2016 A review of Gammaridae (Crustacea: Amphipoda): the family extent, its
evolutionary history, and taxonomic redefinition of genera. *Zoological Journal of the Linnean Society*.
**176**, 323-348. (10.1111/zoj.12318)

MacDonald, K. S., 3rd, Yampolsky, L., Duffy, J. E. 2005 Molecular and morphological evolution
of the amphipod radiation of Lake Baikal. *Molecular Phylogenetics and Evolution*. **35**, 323-343.
(10.1016/j.ympev.2005.01.013)

Lee, C.-W., Nakano, T., Tomikawa, K., Min, G.-S. 2018 The complete mitochondrial genome of
Pseudocrangonyx daejeonensis (Crustacea: Amphipoda: Pseudocrangonyctidae). *Mitochondrial DNA*
*Part B*. **3**, 823--824. (10.1080/23802359.2018.1495116)

Krebes, L., Bastrop, R. 2012 The mitogenome of Gammarus duebeni (Crustacea Amphipoda): A
new gene order and non-neutral sequence evolution of tandem repeats in the control region. *Comp*
*Biochem Physiol Part D Genomics Proteomics*. **7**, 201-211. (10.1016/j.cbd.2012.02.004)

Cormier, A., Wattier, R., Teixeira, M., Rigaud, T., Cordaux, R. 2018 The complete mitochondrial
genome of Gammarus roeselii (Crustacea, Amphipoda): insights into mitogenome plasticity and
evolution. *Hydrobiologia*. 1–14. (10.1007/s10750-018-3578-z)

Sun, S. e., Hui, M., Wang, M., Sha, Z. 2018 The complete mitochondrial genome of the
alvinocaridid shrimp Shinkaicaris leurokolos (Decapoda, Caridea): Insight into the mitochondrial
genetic basis of deep-sea hydrothermal vent adaptation in the shrimp. *Comparative Biochemistry and*
*Physiology - Part D: Genomics and Proteomics*. **25**, 42–52. (10.1016/j.cbd.2017.11.002)

Oliveira, D. C. S. G., Raychoudhury, R., Lavrov, D. V., Werren, J. H. 2008 Rapidly Evolving
Mitochondrial Genome and Directional Selection in Mitochondrial Genes in the Parasitic Wasp
Nasonia (Hymenoptera: Pteromalidae). *Molecular Biology and Evolution*. **25**, 2167-2180.

(10.1093/molbev/msn159)

Xiao, J.-H., Jia, J.-G., Murphy, R. W., Huang, D.-W. 2011 Rapid Evolution of the Mitochondrial
Genome in Chalcidoid Wasps (Hymenoptera: Chalcidoidea) Driven by Parasitic Lifestyles. *PLOS ONE*.
**6**, e26645. (10.1371/journal.pone.0026645)

Carlini, D. B., Fong, D. W. 2017 The transcriptomes of cave and surface populations of *Gammarus*
*minus* (Crustacea: Amphipoda) provide evidence for positive selection on cave downregulated
transcripts. *PLoS ONE*. **12**, e0186173. (10.1371/journal.pone.0186173)

Pons, J., Bauzà-Ribot, M. M., Jaume, D., Juan, C. 2014 Next-generation sequencing, phylogenetic
signal and comparative mitogenomic analyses in Metacrangonyctidae (Amphipoda: Crustacea). *BMC*
*Genomics*. **15**, 566. (10.1186/1471-2164-15-566)

Doublet, V., Ubrig, E., Alioua, A., Bouchon, D., Marcade, I., Marechal-Drouard, L. 2015 Large
gene overlaps and tRNA processing in the compact mitochondrial genome of the crustacean
*Armadillidium vulgare*. *RNA Biol*. **12**, 1159-1168. (10.1080/15476286.2015.1090078)

Zhang, D., Li, W. X., Zou, H., Wu, S. G., Li, M., Jakovlić, I., Zhang, J., Chen, R., Wang, G. T.
2018 Mitochondrial genomes of two diplectanids (Platyhelminthes: Monogenea) expose paraphyly of
the order Dactylogyridea and extensive tRNA gene rearrangements. *Parasites & Vectors*. **11**, 601.
(10.1186/s13071-018-3144-6)

Zhang, D., Zou, H., Wu, S. G., Li, M., Jakovlić, I., Zhang, J., Chen, R., Li, W. X., Wang, G. T.
2018 Three new Diplozoidae mitogenomes expose unusual compositional biases within the Monogenea
class: implications for phylogenetic studies. *BMC evolutionary biology*. **18**, 133.

Gissi, C., Iannelli, F., Pesole, G. 2008 Evolution of the mitochondrial genome of Metazoa as
exemplified by comparison of congeneric species. *Heredity*. **101**, 301-320. (10.1038/hdy.2008.62)

!!! INVALID CITATION !!! [36],

Almeida, D., Maldonado, E., Vasconcelos, V., Antunes, A. 2015 Adaptation of the mitochondrial
genome in cephalopods: Enhancing proton translocation channels and the subunit interactions. *PLoS*
*ONE*. **10**, 1–29. (10.1371/journal.pone.0135405)

Zhang, D., Zou, H., Wu, S. G., Li, M., Jakovlić, I., Zhang, J., Chen, R., Li, W. X., Wang, G. T.
2018 Three new Diplozoidae mitogenomes expose unusual compositional biases within the Monogenea
class: Implications for phylogenetic studies. *BMC Evolutionary Biology*.
(10.1186/s12862-018-1249-3)

Zhang, D., Zou, H., Wu, S. G., Li, M., Jakovlic, I., Zhang, J., Chen, R., Wang, G. T., Li, W. X.
2017 Sequencing of the complete mitochondrial genome of a fish-parasitic flatworm
*Paratetraonchoides inermis* (Platyhelminthes: Monogenea): tRNA gene arrangement reshuffling and
implications for phylogeny. *Parasit Vectors*. **10**, 462. (10.1186/s13071-017-2404-1)

Burland, T. G. 2000 DNASTAR's Lasergene sequence analysis software. In *Methods in Molecular*
*Biology*TM. (ed. ^eds. S. Misener, S. A. Krawetz), pp. 71–91. Totowa, NJ: Humana Press.

Hazkani-Covo, E., Zeller, R. M., Martin, W. 2010 Molecular poltergeists: mitochondrial DNA
copies (numts) in sequenced nuclear genomes. *PLoS Genet*. **6**, e1000834.
(10.1371/journal.pgen.1000834)

Schattner, P., Brooks, A. N., Lowe, T. M. 2005 The tRNAscan-SE, snoscan and snoGPS web
servers for the detection of tRNAs and snoRNAs. *Nucleic Acids Res*. **33**, W686-689.
(10.1093/nar/gki366)

Laslett, D., Canback, B. 2008 ARWEN: a program to detect tRNA genes in metazoan
mitochondrial nucleotide sequences. *Bioinformatics*. **24**, 172-175. (10.1093/bioinformatics/btm573)

Zhang, D., Gao, F., Li, W. X., Jakovlić, I., Zou, H., Zhang, J., Wang, G. T. 2018 PhyloSuite: an
integrated and scalable desktop platform for streamlined molecular sequence data management and
evolutionary phylogenetics studies. *bioRxiv*. 489088. (10.1101/489088)

Lima, N. C. B., Prosdocimi, F. 2017 The heavy strand dilemma of vertebrate mitochondria on
genome sequencing age: number of encoded genes or G + T content? *Mitochondrial DNA Part A*. **29**,
300-302. (10.1080/24701394.2016.1275603)

Lavrov, D. V., Boore, J. L., Brown, W. M. 2000 The complete mitochondrial DNA sequence of the
horseshoe crab *Limulus polyphemus*. *Mol Biol Evol*. **17**, 813-824.
(10.1093/oxfordjournals.molbev.a026360)

Cook, C. E. 2005 The complete mitochondrial genome of the stomatopod crustacean *Squilla* mantis.
*BMC Genomics*. **6**, 105. (10.1186/1471-2164-6-105)

Costello, M. J., Bouchet, P., Boxshall, G., Fauchald, K., Gordon, D., Hoeksema, B. W., Poore, G.
C., van Soest, R. W., Stohr, S., Walter, T. C., *et al.* 2013 Global coordination and standardisation in
marine biodiversity through the World Register of Marine Species (WoRMS) and related databases.
*PLoS One*. **8**, e51629. (10.1371/journal.pone.0051629)

Katoh, K., Standley, D. M. 2013 MAFFT multiple sequence alignment software version 7:
improvements in performance and usability. *Mol Biol Evol*. **30**, 772-780. (10.1093/molbev/mst010)

Lanfear, R., Calcott, B., Ho, S. Y. W., Guindon, S. 2012 PartitionFinder: Combined Selection of
Partitioning Schemes and Substitution Models for Phylogenetic Analyses. *Molecular Biology and*
*Evolution*. **29**, 1695–1701. (10.1093/molbev/mss020)

Nguyen, L. T., Schmidt, H. A., von Haeseler, A., Minh, B. Q. 2015 IQ-TREE: a fast and effective
stochastic algorithm for estimating maximum-likelihood phylogenies. *Molecular Biology and*
*Evolution*. **32**, 268-274. (10.1093/molbev/msu300)

Ronquist, F., Teslenko, M., van der Mark, P., Ayres, D. L., Darling, A., Höhna, S., Larget, B., Liu,
799 L., Suchard, M. A., Huelsenbeck, J. P. 2012 MrBayes 3.2: efficient Bayesian phylogenetic inference
and model choice across a large model space. *Syst Biol*. **61**, 539-542. (10.1093/sysbio/sys029)

Lartillot, N., Brinkmann, H., Philippe, H. 2007 Suppression of long-branch attraction artefacts in
the animal phylogeny using a site-heterogeneous model. *BMC Evolutionary Biology*. **7**, S4.
(10.1186/1471-2148-7-S1-S4)

Talavera, G., Castresana, J. 2007 Improvement of phylogenies after removing divergent and
ambiguously aligned blocks from protein sequence alignments. *Syst Biol*. **56**, 564-577.
(10.1080/10635150701472164)

Kalyaanamoorthy, S., Minh, B. Q., Wong, T. K. F., von Haeseler, A., Jermini, L. S. 2017
ModelFinder: fast model selection for accurate phylogenetic estimates. *Nat Methods*. **14**, 587-589.
(10.1038/nmeth.4285)

Chernomor, O., von Haeseler, A., Minh, B. Q. 2016 Terrace Aware Data Structure for
Phylogenomic Inference from Supermatrices. *Systematic Biology*. **65**, 997-1008.
(10.1093/sysbio/syw037)

Miller, M. A., Pfeiffer, W., Schwartz, T. Year Creating the CIPRES Science Gateway for inference
of large phylogenetic trees. 2010 2010; 2010.

Letunic, I., Bork, P. 2007 Interactive Tree Of Life (iTOL): an online tool for phylogenetic tree
display and annotation. *Bioinformatics*. **23**, 127-128. (10.1093/bioinformatics/btl529)

Librado, P., Rozas, J. 2009 DnaSP v5. *Bioinformatics*. **25**, 1451-1452.

Weaver, S., Shank, S. D., Spielman, S. J., Li, M., Muse, S. V., Kosakovsky Pond, S. L. 2018

Datamonkey 2.0: A Modern Web Application for Characterizing Selective and Other Evolutionary
Processes. *Molecular Biology and Evolution*. **35**, 773–777. (10.1093/molbev/msx335)

Formatted: English (United Kingdom)